# Kondo interaction in FeTe and its potential role in the magnetic order

Younsik Kim[1,2], Min-Seok Kim[3], Dongwook Kim[4], Minjae Kim[5], Minsoo Kim [1,2], Cheng-Maw Cheng [6], Joonyoung Choi [7], Saegyeol Jung[1,2], Donghui Lu [8], Jong Hyuk Kim[9], Soohyun Cho[10], Dongjoon Song[1,2], Dongjin Oh [1,2,16], Li Yu [11,12,13], Young Jai Choi[9], Hyeong-Do Kim[14], Jung Hoon Han[15], Younjung Jo[7], Ji Hoon Shim [4], Jungpil Seo [3], Soonsang Huh [1,2] ✉ & Changyoung Kim [1,2] ✉

Finding $d$-electron heavy fermion states has been an important topic as the diversity in $d$-electron materials can lead to many exotic Kondo effect-related phenomena or new states of matter such as correlation-driven topological Kondo insulator. Yet, obtaining direct spectroscopic evidence for a d-electron heavy fermion system has been elusive to date. Here, we report the observation of Kondo lattice behavior in an antiferromagnetic metal, FeTe, via angle-resolved photoemission spectroscopy, scanning tunneling spectroscopy and transport property measurements. The Kondo lattice behavior is represented by the emergence of a sharp quasiparticle and Fano-type tunneling spectra at low temperatures. The transport property measurements confirm the low-temperature Fermi liquid behavior and reveal successive coherent-incoherent crossover upon increasing temperature. We interpret the Kondo lattice behavior as a result of hybridization between localized Fe $3d_{xy}$ and itinerant Te $5p_z$ orbitals. Our observations strongly suggest unusual cooperation between Kondo lattice behavior and long-range magnetic order.

Coupling between spin and electronic degrees of freedom in condensed matter systems leads to a variety of emergent phenomena such as colossal magnetoresistance, Rashba effect, anomalous Hall effect, and unconventional superconductivity[1-5]. In particular, understanding how the spin and electronic degrees of freedom interact in such systems is the key to elucidating the underlying physical mechanism and can thus be a steppingstone to future practical applications.

One of the canonical fields to study the interplay of these degrees of freedom is heavy-fermion (HF) materials[6,7]. HF states appear as a result of the interaction between itinerant electrons and localized magnetic moments, known as Kondo interaction. Previous

[1]Center for Correlated Electron Systems, Institute for Basic Science, Seoul 08826, Korea. [2]Department of Physics & Astronomy, Seoul National University, Seoul 08826, Korea. [3]Department of Emerging Materials Science, DGIST, Daegu 42988, Korea. [4]Department of Chemistry, Pohang University of Science and Technology (POSTECH), Pohang 37673, Korea. [5]Korea Institute for Advanced Study, Seoul 02455, Korea. [6]National Synchrotron Radiation Research Center, Hsinchu 30076, Taiwan. [7]Department of Physics, Kyungpook National University, Daegu 41566, Korea. [8]Stanford Synchrotron Radiation Light Source, SLAC National Accelerator Laboratory, Menlo Park, CA 94025, USA. [9]Department of Physics, Yonsei University, Seoul 03021, Korea. [10]Center for Excellence in Superconducting Electronics, State Key Laboratory of Functional Materials for Informatics, Shanghai Institute of Microsystem and Information Technology, Chinese Academy of Sciences, 200050 Shanghai, China. [11]Beijing National Laboratory for Condensed Matter Physics and Institute of Physics, Chinese Academy of Sciences, 100190 Beijing, China. [12]School of Physical Sciences, University of Chinese Academy of Sciences, 100049 Beijing, China. [13]Songshan Lake Materials Laboratory, 523808 Dongguan, Guangdong, China. [14]XFEL Beamline Division, Pohang Accelerator Laboratory, Pohang 37673, Korea. [15]Department of Physics, Sungkyunkwan University, Suwon 16419, Korea. [16]Present address: Department of Physics, Massachusetts Institute of Technology, Cambridge, MA 02139, USA. ✉e-mail: sshuhss@gmail.com; changyoung@snu.ac.kr

experimental/theoretical studies show most of the HF materials are f-electron systems[3,6,7]. It was only recently proposed that d-electron systems can also host HF states via Kondo interactions[8–11]. HF states in d-electron materials are especially important due to the possibility that the diversity of d-electron systems may result in exotic Kondo interaction-related phenomena, such as topological Kondo insulating state[12] or cooperation between Kondo lattice behavior and long-range magnetism[13]. Thus, the novelty calls for new studies to find HF in d-electron material groups.

FeTe can be a candidate material to observe d-electron HF states. Its electron correlation is the strongest among the iron-based superconductors (IBSCs)[14]. The magnetic ground state is known to be bicollinear antiferromagnetism (BAFM) with a large magnetic moment of 2.1 $\mu_B$, implying its local nature of the magnetism[14]. The Sommerfeld coefficient of FeTe is reported to be 31.4 mJ/(K$^2$ mol), indicating a heavy effective mass of the system[15]. This value is much larger than that of other iron chalcogenides; FeS and FeSe for instance have 3.8 and 6.9 mJ/(K$^2$ mol), respectively[16,17].

In addition to these HF-related properties, other transport properties suggest the existence of strong spin-electron interaction. The temperature-dependent resistivity exhibits a drastic change at the Néel temperature ($T_N$). It shows an insulating behavior above $T_N$, but a metallic behavior below $T_N$[18]. The aforementioned properties of FeTe imply that the local magnetic moment significantly affects the electronic structure. Thus, electronic structure studies on the HF state of FeTe can unveil its origin and how it couples with magnetism.

Here, we report on a comprehensive study on FeTe using angle-resolved photoemission spectroscopy (ARPES), transport property measurements, and scanning tunneling spectroscopy (STS). We observe a hallmark of an HF behavior in ARPES spectra: a sharp quasiparticle peak (QP) near the Γ point and its strong temperature dependence. The observed QP is attributed to Kondo hybridization between Fe 3d$_{xy}$ and Te 5p$_z$. The Kondo hybridization scenario is further supported by STS results, showing the Fano line shape and narrow hybridization gap. In this picture, the recovery of metallic behavior in the low-temperature region is due to the emergence of the strong QP

around the Γ point. We also conducted a Heisenberg model calculation, suggesting the Kondo interaction may be responsible for the emergence of BAFM in FeTe. These results provide a unified perspective that the Kondo interaction determines the exotic physical and magnetic properties in FeTe.

## Results

### Transport properties

FeTe has the simplest crystal structure among the IBSCs as shown in Fig. 1a. Compared to other similar iron chalcogenide systems of FeSe and FeS, FeTe has a distinctive bonding angle value $\theta$ shown in Fig. 1a. More specifically, Te atom is pushed away from the Fe plane due to its large atomic size and, as a result, FeTe has a small $\theta$ value[19]. This aspect of the crystal structure leads to localization of the Fe 3d$_{xy}$ band as the d$_{xy}$ orbital is confined in the Fe plane[13,19]. A recent ARPES study showed a complete loss of coherent spectral weight in the d$_{xy}$ band in FeTe, indicating a strong localization in the band[14,20]. The magnetic ground state of FeTe is bicollinear antiferromagnetism (BAFM) as shown in Fig. 1b below a Néel temperature of near 70 K[18]. It is noteworthy that among IBSCs, only FeTe exhibits BAFM. The ordering vector of BAFM in FeTe is (π/2, π/2) (1-Fe unit cell) while that of conventional AFM shown on other IBSCs is (π, 0)[21].

Transport properties show a close relationship with magnetic properties. The temperature-dependent resistivity in Fig. 1c shows insulating behavior above $T_N$. We find the temperature dependence follows a logarithmic behavior of -ln(T). On the other hand, it abruptly recovers a metallic behavior below $T_N$. More specifically, it shows a Fermi liquid behavior below 15 K with a T$^2$ dependence resistivity, and a T-linear behavior between 30 K and 70 K. These T-dependent behaviors indicate the existence of coherent-incoherent crossover around 15 K (see the inset of Fig. 1c and Supplementary Note 1). It is also noteworthy that the resistivity shows a minimum at around 2.2 K (see Supplementary Note 2 for the corresponding data and discussion). The Hall coefficient, as well as the resistivity, shows a drastic change at $T_N$. The Hall coefficient changes hole dominant (T > $T_N$) to electron dominant (T < $T_N$) at $T_N$ as can be seen in Fig. 1d. The crossover

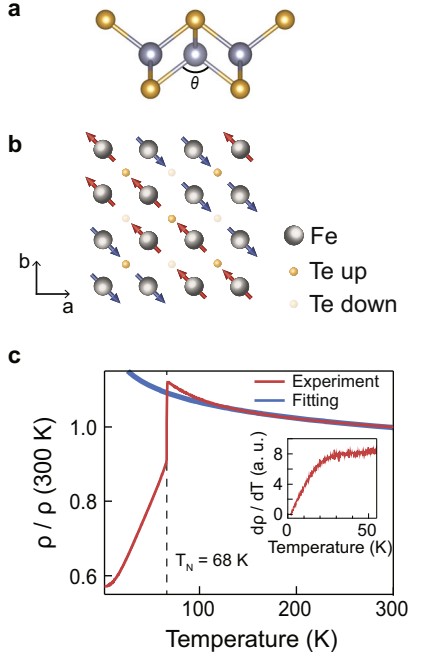

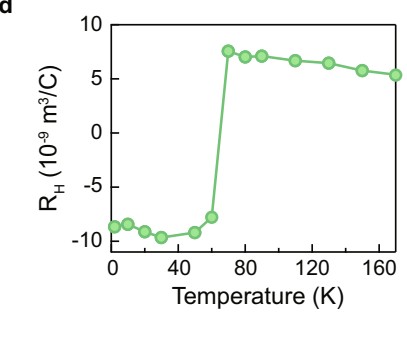

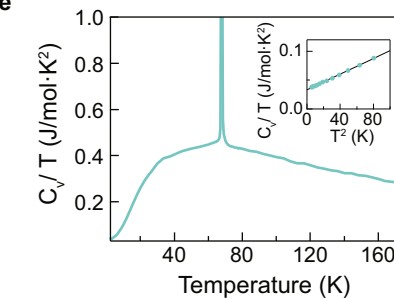

**Fig. 1 | Crystal structure and transport results of FeTe. a** Crystal structure of FeTe. **b** Spin configuration of bicollinear antiferromagnetic (BAFM) state in FeTe. **c** Temperature-dependent resistivity. The red curve is the experimental data, while the blue curve is the fitting result of the logarithmic function ($a + b\log(T)$) of the

data between 120 and 300 K. Inset shows the temperature-derivative of the resistivity. **d** Temperature-dependent Hall coefficient. **e** Temperature-dependent C$_v$/T. Inset shows C$_v$/T vs T$^2$ plot in the low-temperature region. The black solid line in the inset is the fit result of C$_v$/T = γ + βT$^2$. Source data are provided as a Source Data file.

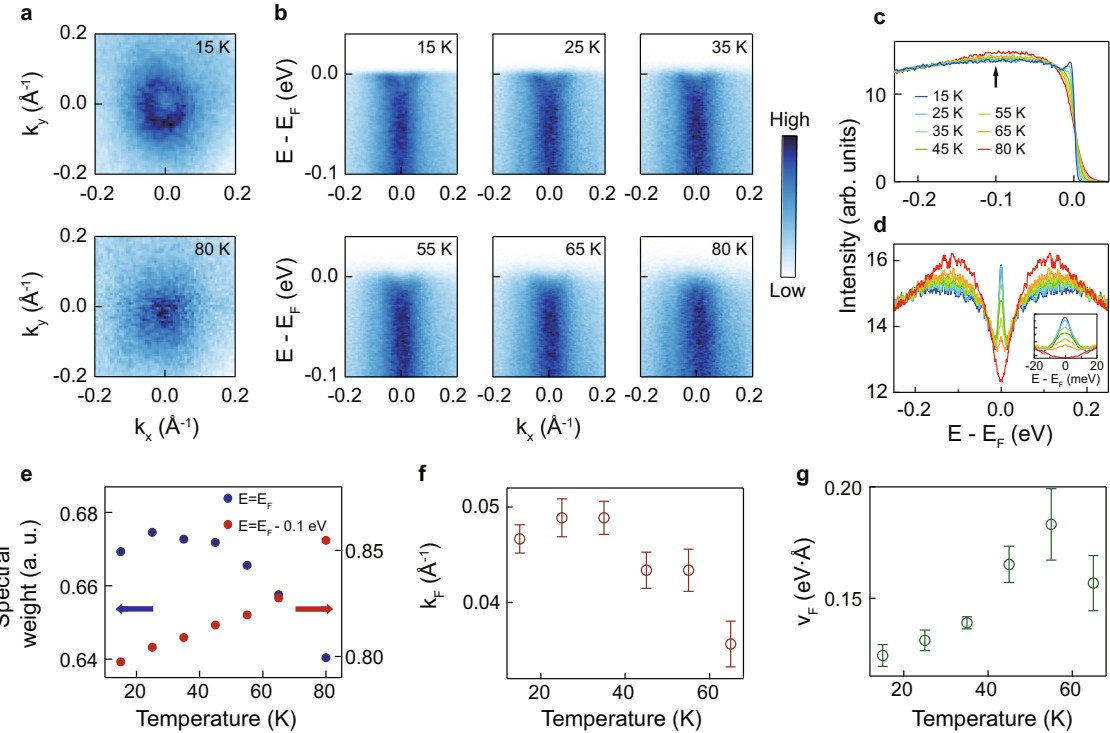

**Fig. 2 | Electronic structure of FeTe. a** Fermi surface (FS) maps from high-resolution laser ARPES measurements, obtained at 15 and 80 K. **b** Temperature-dependent high symmetry cuts along the Γ'-X' direction. ARPES data were taken with 11 eV photons. **c** Energy distribution curves (EDCs) integrated within a certain momentum range ($k_x^2 + k_y^2 < (0.15\ \text{Å}^{-1})^2$). The EDCs are normalized with the integrated intensity from an energy window of $-0.25\ \text{eV} < E - E_F < -0.2\ \text{eV}$.

**d** Symmetrized EDCs of (**c**). Inset: enlarged view of EDCs near the Fermi level. **e** Temperature-dependent spectral weight at $E = E_F$ and $E = E_F - 0.1\ \text{eV}$. **f, g** Temperature-dependent Fermi momentum ($k_F$) and Fermi velocity ($v_F$), respectively, obtained from momentum distribution curve (MDC) analysis. Errors bars in (**f, g**) represent the fitting errors of Fermi momentum and Fermi velocity, respectively. Source data are provided as a Source Data file.

behavior seen in the resistivity data can be also found in the heat capacity data in Fig. 1e; $C_v/T$ deviates from $T^2$ behavior around 15 K (see Supplementary Note 1 for the determination of the deviation temperature). Further analysis shows that the Sommerfeld coefficient extracted from the heat capacity is 33.4 mJ/mol K$^2$ (see the inset of Fig. 1e). It is much larger than that of other iron chalcogenides. For instance, it is 3.8 and 6.9 mJ/mol K$^2$ for FeS and FeSe, respectively[16,17].

## Electronic structures

We turn our attention to the electronic structure of FeTe. High-resolution laser ARPES experiments were performed to track the temperature-dependent evolution of the electronic structure. The Fermi surfaces (FSs) near the Γ' point (corresponds to $k_z \approx 0.5\ \pi/c$ where $c$ is the lattice constant along the $z$ direction; see "Methods" for details) shown in Fig. 2a exhibit significant temperature dependence as the temperature decreases from 80 to 15 K. A single circular FS pocket is clearly observed at 15 K while it becomes a blob at 80 K. Evolution of the electronic structure can also be seen in the high symmetry cuts along the $k_x$-direction shown in Fig. 2b. It is revealed that the FS pocket observed at 15 K in Fig. 2a comes from an electron band. As the temperature increases, the electron band tends to be broadened and vanishes abruptly at 80 K.

This observed temperature dependence of the band can be more clearly seen in the temperature-dependent energy distribution curves (EDCs) plotted in Fig. 2c. A clear QP is observed at the lowest temperature, which comes from the electron band mentioned above. Upon increasing temperature, the QP is gradually suppressed while the spectral weight of the hump centered at $-0.1$ eV, indicated by an arrow in Fig. 2c, gradually increases. Such spectral weight transfer behavior is more pronounced in symmetrized EDCs in Fig. 2d. Analysis of the spectral weight transfer behavior is depicted in Fig. 2e. It clearly shows

that the lost QP spectral weight is transferred to the 0.1 eV hump, demonstrating that the observed temperature dependence is intrinsic. It is also noteworthy that the full width at half maximum (FWHM) of the QP obtained from a Lorentzian fitting is 7.9 meV as can be seen in the inset of Fig. 2d, implying remarkable heavy mass and long quasiparticle lifetime of the band.

Additional band-fitting analyses provide more information about the temperature-dependent evolution of the band. We extract the Fermi momentum ($k_F$) and Fermi velocity ($v_F$) using momentum distribution curve (MDC) analysis as depicted in Fig. 2f, g, respectively. Temperature-dependent $k_F$ value shows that the FS pocket size tends to enlarge upon cooling. Meanwhile, $v_F$ of the electron band decreases with the temperature. From these results, we can infer that the temperature evolution of the $k_F$ and $v_F$ did not result from a simple chemical potential shift. The origin of the evolution will be discussed below.

The photon energy-dependent ARPES result gives further insights into the origin of the band. As can be seen in Fig. 3, the electron band which is clearly visible at 11 eV has a strong $k_z$ dispersion. As the photon energy increases, the band shifts to the higher binding energy side, and its energy scale becomes more than 0.5 eV. Considering FeTe is in the strongly correlated limit, a bandwidth of 0.5 eV far surpasses that of Fe 3d bands[19]. In addition, the photoionization cross section of Te 5p orbital is much larger than that of Fe 3d orbital at 11 eV[22]. Thus, the band observed at 11 eV is likely to be mostly from Te $p_z$ orbital. We note that similar $k_z$ dispersion behavior was also reported for FeTe$_{0.55}$Se$_{0.45}$[23]. Polarization- and experimental geometry-dependent ARPES measurements also confirm the $p_z$ character of the band (see Supplementary Note 10).

Considering the large dispersion of the $p_z$ band away from $E_F$ as shown in Fig. 3, the sharp QP near $E_F$ implies that the band undergoes a strong modulation. Two scenarios may be considered for the

**a**

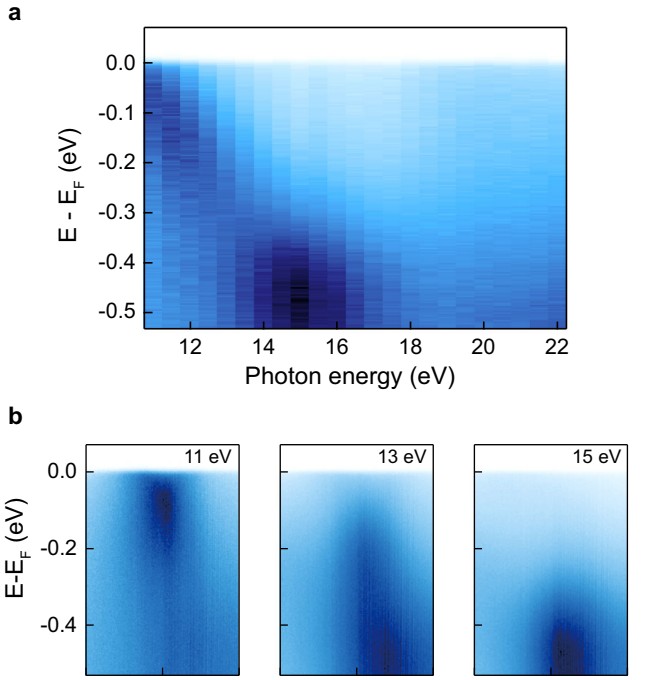

**b**

**Fig. 3 | Photon energy-dependent electronic structure. a** Photon energy-dependent electronic structure near the Γ point. **b** Photon energy-dependent high symmetry cuts along the Γ-X direction, obtained using 11, 13, 15 eV photon. Source data are provided as a Source Data file.

modulation: (i) electron-bosonic mode coupling and (ii) Kondo hybridization between the itinerant and localized bands. It was claimed in a previous ARPES study on FeTe that the feature is a result of strong electron-phonon coupling, namely a polaronic behavior[24]. However, such a scenario may not explain the enlargement of the Fermi surface at low temperatures in Fig. 2f since an electron-boson coupling should conserve the $k_F$. Alternatively, one can consider a Kondo hybridization scenario which should also show a mass enhancement at low temperatures and strong temperature dependence of the QP. Therefore, it is highly desirable to have an alternative way to discern the two scenarios.

## Fano line shape and hybridization gap

Whether the strong renormalization of the dispersion near $E_F$ is due to Kondo hybridization or not may be determined based on tunneling spectra. Shown in Fig. 4 are STS data at 4.3 and 80 K. A wide energy range scan at 4.3 K depicted in Fig. 4a shows an asymmetric spectrum. The spectrum is found to be well-fitted with a Fano line shape as illustrated in the figure. It is well-known that tunneling spectra from a Kondo singlet state should exhibit a Fano-type resonance[13,25]. The Fano fit shown as blue circles in Fig. 4a gives a Fano line width (Γ value) of 24.1 meV, which corresponds to the Kondo temperature of about 280 K. Furthermore, a closer look of the data over a narrow energy range around $E_F$ plotted in Fig. 4b shows a gap feature that is consistent with a gap expected for a Kondo hybridization scenario. We subtract the smoothly varying background from the data and plot it in the inset. The subtracted data shows a gap with a size of about 7 meV as seen in Fig. 4b. In addition, it is seen that the gap feature is slightly shifted to the unoccupied side. Plotted in Fig. 4c, d are d$I$/d$V$ spectra taken at 80 K, above $T_N$. The two spectra are taken over the same energy ranges as the 4.3 K data. The Kondo-related features are expected to disappear at high temperatures, which are indeed seen in

**a**

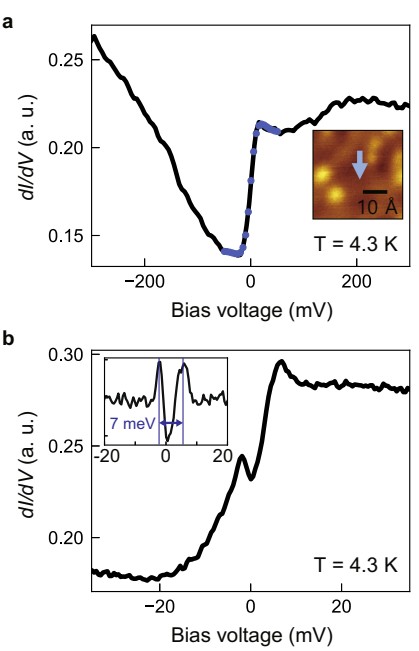

**c**

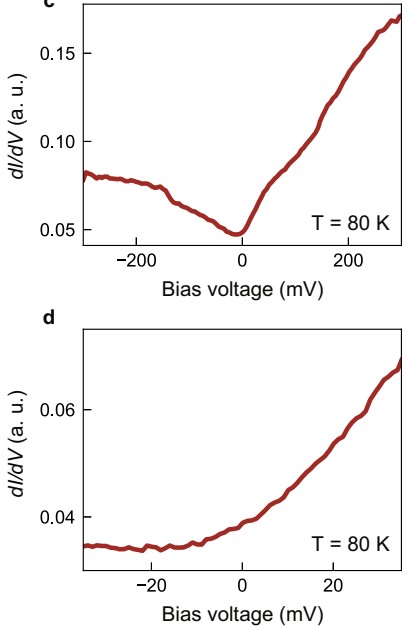

**b**

**d**

**Fig. 4 | STS results on FeTe. a** Differential conductance (d$I$/d$V$) spectrum measured on FeTe surface at 4.3 K. The blue circles represent the Fano fitting of the Kondo resonance (see Supplementary Note 4 for the fitting parameters). The inset shows the position where the spectrum is taken. $V_{bias}$ = −300 mV, $I$ = 100 pA, and lock-in modulation $V_{mod}$ = 5 mV$_{pp}$. **b** d$I$/d$V$ spectrum enlarged around the Fermi

energy. The inset is the spectrum after subtracting the smoothly varying background. $V_{bias}$ = −40 mV, $I$ = 100 pA and $V_{mod}$ = 500 μV$_{pp}$. **c** d$I$/d$V$ spectra measured at 80 K. $V_{bias}$ = −300 mV, $I$ = 50 pA and $V_{mod}$ = 5 mV$_{pp}$. **d** Zoomed-in d$I$/d$V$ spectrum. $V_{bias}$ = −40 mV, $I$ = 50 pA and $V_{mod}$ = 500 μV$_{pp}$. Source data are provided as a Source Data file.

the high-temperature data in Fig. 4c, d; the Fano behavior is weakened and the hybridization gap has disappeared. Therefore, these observations—Fano behavior and narrow gap near $E_F$—are clear signs of Kondo hybridization, confirming that FeTe exhibits Kondo hybridization below $T_N$.

## Discussion

Fully considering our comprehensive data, we argue that the electron band that emerges below $T_N$ is a result of a Kondo hybridization between the itinerant $p_z$ and localized $d_{xy}$ bands. The argument is based on the fact that only the $d_{xy}$ orbital of FeTe is in a localized state, which is a prerequisite for the Kondo effect[6,14,19]. Density functional theory calculations also confirm the band we measured in ARPES has Te $p_z$ and Fe $d_{xy}$ orbital characters (see Supplementary Note 8). Here, it is also noteworthy that the appearance of the coherence peak may be accounted for within the coherence-incoherence crossover picture in Hund's metal[26,27] as observed in some of the iron-based superconductors[28–30]. However, the Kondo hybridization picture is needed to explain the other aspects of the experimental results. Indeed, recent theoretical work proposed that the interorbital hopping in the orbital-selective Mott phase can develop a narrow quasiparticle peak near the Fermi level[31]. In this perspective, our work emphasizes the role of interorbital coupling. When the system enters the BAFM state, the $p_z$ and $d_{xy}$ bands start to Kondo hybridize as illustrated in Fig. 5; the strongly dispersive $p_z$ band along $k_z$ direction crosses the localized $d_{xy}$ band, resulting in a Kondo hybridization and heavy electron band. The correlation between Kondo hybridization and BAFM is discussed later. Based on known band dispersions, we simulate the band structure with a finite hybridization between the $p_z$ and $d_{xy}$ band. The simulated band structures projected onto the (001) surface in Fig. 5e, f well coincide with ARPES results shown in Fig. 2b at the temperature of 80 K and 15 K, respectively. In addition, the narrow gap in the unoccupied side at low temperature and its disappearance at high temperature in the STS data directly support the band diagram illustrated in Fig. 5f, e, respectively. The details of the simulation are described in the Materials and Methods section. The Kondo hybridization scenario is further supported by previous inelastic neutron scattering measurements on FeTe: the study reported that the local magnetic moment of FeTe is $S = 1$ at 10 K but it unexpectedly grows to $S = 3/2$ at 300 K, suggesting low-temperature Kondo screening of the local moments by itinerant electrons[32]. Note that the $d_{xy}$ band is not visible near the Fermi level since $d_{xy}$ band is strongly localized, and thus its spectral weight near the Fermi level is mostly transferred to the high binding energy region, and the photoionization cross section of Te 5p orbitals far surpass that of Fe 3d orbitals at 11 eV photon[20,23].

The observed heavy electron band resulting from Kondo hybridization can address the unique transport properties of FeTe: (i) recovery of metallic behavior below $T_N$, (ii) sudden sign change in the Hall conductivity at $T_N$, and (iii) emergent Fermi liquid behavior at low temperature. First, the recovery of metallic behavior can be understood through the emergence of the sharp and strong QP at the Fermi level near the Γ point at $T_N$; the transport properties are dominated by the QP. The emergence of the electron QP below $T_N$ can also explain the sign change in the Hall conductivity, from hole dominant ($T > T_N$) to electron dominant ($T < T_N$). A previous study reported that recovery of the metallic behavior and Hall coefficient change may be related to the formation of pseudogap near the Brillouin zone corner[33]. However, their observation is not enough to explain the abrupt change in the resistivity and Hall conductivity. It is also noteworthy that such a strong QP and its strong temperature dependence are only observed at the Γ point (see Supplementary Note 3 for the temperature-dependent ARPES results on the X point pocket.). Thus, we believe the FS near the Γ point, which exhibits a sudden change at $T_N$, dominates transport properties. Finally, the sharp QP bandwidth of 7.9 meV indicates a long quasiparticle lifetime, indicating that FeTe is in a Fermi liquid regime at

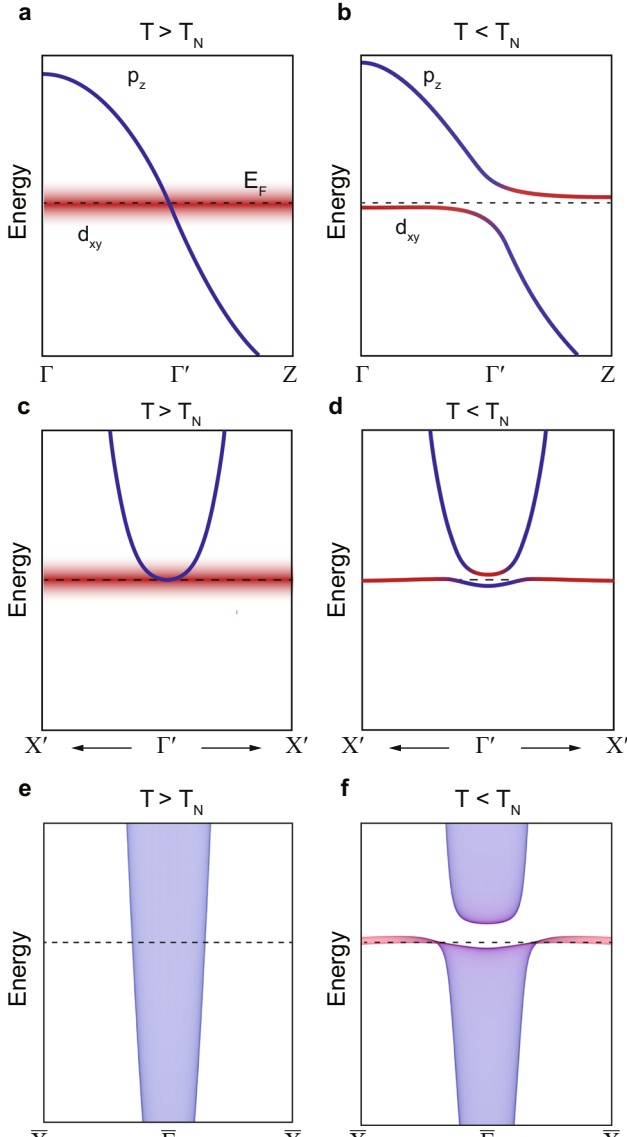

**Fig. 5 | Schematic of the Kondo hybridization scenario. a, b** Band structure of FeTe along the Γ-Z direction (out-of-plane) above and below $T_N$, respectively. **c, d** Band structure of FeTe along the Γ-X direction (in-plane) above and below $T_N$, respectively. **e, f** Simulated band structure projected onto the (001) surface along the Γ-X direction (in-plane) above and below $T_N$, respectively. Blue bands denote $p_z$ orbital, and red bands denote $d_{xy}$ orbital.

low temperatures. This observation is consistent with the unique transport results and enhanced Sommerfeld coefficient of FeTe. We note that recent ARPES and STS studies on CeRh$_2$Si$_2$ and SmB$_6$ reported significantly different Kondo properties at the surface and in the bulk[34–36]. In such cases, considering the surface sensitivity of ARPES and STS, the Kondo-related properties of FeTe observed via ARPES and STS can be different from those of transport measurements. However, the crystal structure of FeTe is quasi-two-dimensional, which is distinct from CeRh$_2$Si$_2$ and SmB$_6$[20]. This feature might be the reason for the consistency in the Kondo properties of FeTe observed by ARPES and transport measurements.

The overall temperature dependence of electronic structures and transport properties are well explained within the Kondo lattice scenario. In the paramagnetic (PM) state, FeTe is in the Kondo scattering regime, consistent with the logarithmic resistivity[37,38] and estimated Kondo temperature from Fano line width (see Supplementary Note 4

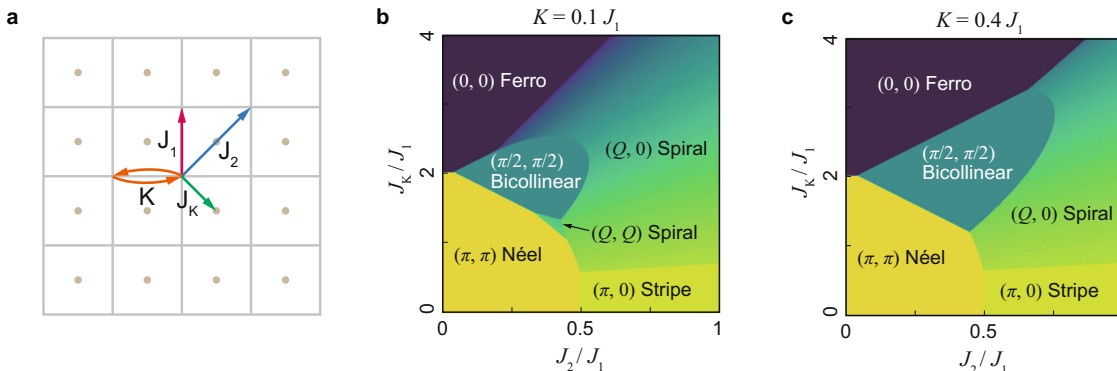

**Fig. 6 | Magnetic phase diagram of FeTe from Heisenberg model. a** Definition of the Heisenberg model parameters. Gray solid lines denote the prime square lattice, whereas brown dots denote the sublattice. $J_1$ and $J_2$ are nearest-neighbor (NN) and next nearest-neighbor (NNN) exchange interactions, respectively, on the prime lattice. $J_K$ denotes NN exchange interaction between the prime lattice and sublattice. $K$ is the NN biquadratic exchange interaction. **b, c** Magnetic phase diagram calculated from the model Hamiltonian (Eq. (1)) with $K = 0.1$ and 0.4, respectively.

for detailed parameters). From the electronic structure point of view, the strong scattering in the Kondo scattering regime results in the breakdown of a well-defined quasiparticle, which in turn leads to loss of spectral weight and its transfer to a higher binding energy region[37,39]. Thus, the hump structure is the incoherent counterpart of the QP, supported by the spectral weight transfer as shown in Fig. 2e. The broadened but persistent Fano line shape at 80 K also indicates the system is still in the Kondo scattering regime, while strongly suppressed coherency above $T_N$ leads to the loss of the QP. On the other hand, when the system enters the BAFM state, low-temperature behaviors of a Kondo lattice emerge: a sharp quasiparticle peak in the electronic structure induced by Kondo hybridization[35,37] as well as a Fermi liquid behavior ($T^2$ dependence) at low temperature followed by a coherent-incoherent crossover in resistivity[8,13,37]. Based on these facts, we may address the unique feature of the Kondo lattice behavior in FeTe; low-temperature Kondo lattice behaviors in FeTe suddenly set in at the onset of BAFM as evidenced by the abrupt drop in the resistivity and sudden emergence of QP at $T_N$. This drastic shift of the system to the low-temperature Kondo lattice regime at the onset of the BAFM suggests a possible positive correlation between BAFM and Kondo lattice behavior in FeTe.

To reveal the underlying mechanism of the positive correlation between BAFM and Kondo lattice behavior in FeTe, we conducted a Heisenberg model calculation with an additional Fe–Te exchange interaction. Based on the established two-neighbor Heisenberg model with the biquadratic term ($J_1$–$J_2$–$K$ model) on a prime square lattice[40-43], we additionally introduce a centered sublattice as shown in Fig. 5a to take into account the Fe–Te interaction (defined as $J_K$ hereafter). We define the $J_1$–$J_2$–$J_K$–$K$ model on the combined lattice as

$$H = J_1 \sum_{<i,j>} \vec{S}_i \cdot \vec{S}_j - K \sum_{<i,j>} \left( \vec{S}_i \cdot \vec{S}_j \right)^2 + J_2 \sum_{\ll i,j \gg} \vec{S}_i \cdot \vec{S}_j + J_K \sum_{<i,k>} \vec{S}_i \cdot \vec{S}_k \quad (1)$$

where $J_1$ and $J_2$ are nearest-neighbor (NN) and next nearest-neighbor (NNN) exchange interactions on the prime lattice, respectively, and $K$ is the NN biquadratic exchange interaction, while $J_K$ is the NN interaction between prime lattice and sublattice as described in Fig. 6a. $i$ and $j$ are indices for the prime lattice, and $k$ is the sublattice index.

We solved the $J_1$–$J_2$–$J_K$–$K$ model for various $K$ values and obtained the corresponding magnetic phase diagram in Fig. 6b, c. For a small $J_K$, the model well reproduces ($\pi$, 0) stripe phase in iron pnictides. As $J_K$ grows, ($\pi/2$, $\pi/2$) BAFM starts to be stabilized and spans the phase diagram over a wide range of $K$ (see Supplementary Fig. 7 for an extended phase diagram.). Within the $J_K$-induced BAFM scenario, the sublattice (Te atom for FeTe) should be also spin-polarized

accordingly. We note that previous spin-polarized scanning tunneling microscopy measurements on FeTe revealed that Te atoms are also spin-polarized in the BAFM state[44]. These results suggest that $J_K$, an exchange interaction between Fe and Te, may play a crucial role in stabilizing the BAFM in FeTe. This $J_K$-induced BAFM scenario thus explains the positive correlation between Kondo lattice behavior and BAFM since the Kondo lattice behavior and BAFM share the same origin, $J_K$. The positive correlation between long-range magnetism and Kondo lattice state is reminiscent of the underscreened Kondo lattice model in UTe and UCu$_{0.9}$Sb$_2$, where a local magnetic moment of S = 1 is not fully screened by itinerant electrons[43,44]. Likewise, the local moment of S = 3/2 in FeTe at 300 K is not fully screened, resulting in a residual local moment of S = 1 at 10 K[32], suggesting a possible analogy with the underscreened Kondo lattice model[45,46].

We find the $J_1$–$J_2$–$J_K$–$K$ model has further implications. It was previously reported that an unexpected ferromagnetic (FM) state emerges under hydrostatic pressure[47]. A transition from BAFM to FM occurs in our calculated magnetic phase diagram if $J_K$ is further increased. Note that previously proposed Heisenberg models had to employ the third nearest-neighbor exchange interaction ($J_3$) to account for the BAFM in FeTe, but could not predict the FM phase[42,43]. In other words, the inclusion of $J_K$ may be the key to understanding the magnetic order in FeTe.

Recently, there have been numerous studies reporting that orbital selectiveness is a prominent ingredient to make physics diverse in correlated d-electron multiorbital systems[14,28,29,48-50]. In particular, while the orbital-selective Mott phase itself is an intriguing phenomenon, another important aspect is that materials with orbital-selective Mott phase are vulnerable to Kondo hybridization and thus may result in a new type of HF state[8-10]. We thus suppose that the local magnetic moment formed in the orbital-selective Mott phase critically affects the physical and magnetic properties of FeTe via Kondo interaction[14]. Our results shed light on the role of the local magnetic moments in correlated d-electron multiorbital systems.

## Methods

### Sample growth and characterization

Single crystals of FeTe were synthesized using a modified Bridgman method[51]. Stoichiometric iron (99.99%) and tellurium (99.999%) were sealed into an evacuated quartz tube and placed in a two-zone furnace. The hot (cold)-zone of the furnace was set to be 1070 (970) °C and slowly cooled down to 570 (470) °C at a rate of 2 °C/h. The estimated excess iron concentration of Fe$_{1+y}$Te is in the range between 0.08 and 0.12, which is determined from STM topography and inductively coupled plasma measurements.

## ARPES measurements

High-resolution ARPES measurements were performed with a home lab-based laser ARPES system equipped with a 10.897 eV laser (UV-2 from Lumeras) and a time-of-flight analyzer (ARTOF 10k from Scienta Omicron)[52]. The $k_z$ of the $\Gamma'$ point is approximately determined to be 0.5 $\pi/c$, with the photon energy of 11 eV and inner potential of 13 eV[53]. Photon energy-dependent ARPES measurements were performed at BL-21B1 of the National Synchrotron Radiation Research Center (NSRRC). All ARPES measurements were conducted with p-polarized light. Overall energy resolution for the laser ARPES and photon energy-dependent ARPES measurements was set to be 2 and 14 meV, respectively. The temperature-dependent measurements were conducted upon cooling, starting from 80 K. The photon energy-dependent measurements were conducted at 15 K.

## Transport measurements

The resistivity and heat capacity measurements were carried out with a Physical Property Measurement System (PPMS from Quantum Design). The resistivity and Hall coefficient measurement was conducted in a standard 4-probe and Hall bar geometry, respectively.

## STM measurements

STM experiments have been performed using a home-built low-temperature STM operating at 4.3 K or 80 K. The FeTe single crystal pre-cooled to 15 K was cleaved in the ultrahigh vacuum condition. The cleaved FeTe sample was immediately inserted into the STM head. A PtIr tip is used for the measurements, and the tip quality is checked by the surface interference pattern on Cu(111). To acquire d$I$/d$V$ spectra, a standard lock-in technique was used with a modulation frequency of $f = 718$ Hz.

## Band structure simulation

The band structure simulation with a toy model is conducted to simulate ARPES results with finite $k_z$ broadening where a strongly $k_z$-dispersive band is hybridized with a localized band. The simulation is based on a two-band model with a finite hybridization. The Hamiltonian is defined as

$$H = \begin{pmatrix} E_p(\vec{k}) & \Delta \\ \Delta & E_d(\vec{k}) \end{pmatrix},$$

where

$$E_p(\vec{k}) = 5t\left(\frac{k_x}{\pi}\right)^2 + 100t\cos(k_z) - \mu,$$

$$E_d(\vec{k}) = -\frac{t}{200}\left(\frac{k_x}{\pi}\right)^2 - t\cos(k_z) - \mu,$$

$$\Delta = 10t.$$

$t$ is the energy scale of the hopping parameter, and µ is the chemical potential of the system which is set arbitrarily. The basis of each axis is p, d orbitals, respectively. The in-plane dispersion is defined as parabolic and out-of-plane dispersion is defined as a cosine function. The dispersion parameter is based on the DFT calculation and ARPES results on FeTe$_{1-x}$Se$_x$[20,22–24]. The diagonalized band structures are projected onto the (001) surface and plotted in Fig. 5f. For Fig. 5e, only $E_p(\vec{k})$ is plotted to simulate the ARPES data at 80 K where hybridization does not occur. The blue and red intensity in Fig. 5 denotes the orbital character of p$_z$ and d$_{xy}$, respectively.

## Data availability

Source data are available at https://doi.org/10.6084/m9.figshare.23538054. Other data that support the findings of this study are available from the corresponding author upon reasonable request.

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

## Acknowledgements

The authors appreciate valuable discussion with T. Tohyama, Y. Bang, A. Go, and S.-S.B. Lee. This work was supported by the Institute for Basic Science in Korea (Grant No. IBS-R009-G2) and the National Research Foundation of Korea (NRF) grant funded by the Korea government (MSIT). (No. 2022R1A3B1077234). The work at Yonsei University was supported by the National Research Foundation of Korea (NRF) (grant numbers NRF-2017R1A5A1014862 (SRC program: vdWMRC center), and NRF-2022R1A2C1006740). Minjae. K was supported by KIAS Individual Grants(CG083501). The STM work was supported by National Research Foundation of Korea (NRF) Grants (No. 2020R1A2C2102838). This work was supported at NSRRC by Grant No. 110-2112-M-213 -016 of the National Science and Technology Council (NSTC), Taiwan. JHS was supported by the National Research Foundation of Korea (NRF) funded by the Ministry of Science and ICT (NRF-2022M3C1A3091988).

## Author contributions

Y.K., S.H., and C.K. conceived the work. Y.K. and S.H. grew single crystals of FeTe. Y.K., Minsoo. K., S.H., S.J., D.S., and S.C. performed ARPES measurements with support from C.-M.C. and D.L. Y.K. built laser ARPES system with support from L.Y. Y.K., J.C., D.O., S.H., J.H.K., Y.J.C., and Y.J. performed transport property measurements. M.-S.K. and J.S. performed STS measurements and analyzed data. Y.K. analyzed the ARPES and transport data. H.-D.K. assisted with the interpretation of results. Y.K. performed Heisenberg model calculations with support from J.H.H. D.K., Minjae. K., and J.H.S. performed electronic structure calculations. Y.K., S.H., and C.K. wrote the manuscript with contributions from all authors. All the authors discussed the results and commented on the paper.

## Competing interests

The authors declare no competing interests.
