## [Peer Review File · Nature Communications]

REVIEWER COMMENTS

Reviewer #1 (Remarks to the Author):

The manuscript by Younsik Kim et al. presents the results of investigations of FeTe system by means of transport measurements, laser-based ARPES, STM/STS measurements and modelling. The most essential result is the observation of Kondo lattice behavior in FeTe and its cooperation with long-range magnetic order.

In general, I agree with the interpretation of the obtained data. However, I believe that the manuscript, in its actual form, is not complete. I have the following points, which should be discussed and addressed in the revised draft.

(1). The authors discuss the spectral pattern and its temperature behavior only in the vicinity of the Gamma point. It is unclear why only this region of BZ was considered. I fully understand that laser-based PE allows probing mainly this region. However, from the PRL 110, 037003 (2013), it looks so that the area around other regions like close to the X-point could be of interest as well. Please, make a comment on that.

(2). The authors say that "...The Kondo lattice behavior as a result of hybridization between localized Fe 3dxy and itinerant Te 5pz orbitals." I did not find that this statement is supported neither from experiment nor from theory.

(3). I should also say that the absence of the results from ab-initio calculations is a weak point of the paper, which is considered for publication in NC.

Recently, it was shown that DMFT and DFT calculations are quite useful for unveiling the Kondo lattice properties in f-materials:

<https://www.pnas.org/doi/epdf/10.1073/pnas.2001778117>

for prediction and observation of Rashba SOC phenomena in f-systems:

<https://onlinelibrary.wiley.com/doi/10.1002/aelm.202100768>

(4). I have to say that ARPES data were accurately obtained, treated and presented. This is seen from figure 2c where the correctly applied normalization allows to detect the QP peak, “hump” structure and their temperature evolution.

I suggest to extend the discussion about the “hump” structure. It is still unclear where it comes from and what is its origin.

(5). Recently, it was shown that the Kondo lattice properties at the surface and in the bulk could be strongly and unexpectedly different:

<https://www.nature.com/articles/s41535-020-00273-7>

<https://www.nature.com/articles/ncomms11029>

In this respect, what the authors could say concerning FeTe.

In conclusion, I think these results could be published in Nature Communications, after answering the questions and considering the suggestions listed above.

Reviewer #3 (Remarks to the Author):

In this manuscript, the authors reported their angle-resolved photoemission spectroscopy (ARPES), scanning tunneling spectroscopy and transport measurements on FeTe single crystal. They found sharp quasiparticle peak and Fano-type tunneling spectra at low temperatures whereas upon increasing temperature, the spectra weight at the Fermi level decreases and spectra undergoes coherent-incoherent crossover. The authors argued that these observations are manifestations of Kondo lattice behavior which is resulted from the hybridization between the localized Fe 3d_{xy} and itinerant Te 5p_z orbitals.

Coherence–incoherence crossover in the normal state of iron-based superconductors was first predicted back in 2009 by Haule and Kotliar (NJP 11, 025021). The coherence-incoherence crossover of FeTe was predicted in 2012 by Yin, Haule and Kotliar (PRB 86, 195141). The temperature evolution of the spectra (ARPES) was shown for FeTe in particular. These and other studies show that Hund's rule coupling plays the dominating role. The coherence-incoherence crossover was observed experimentally in several iron-based superconductors including $AxFe_{2-y}Se_2$ ($A= K, Rb$) (PRL 110, 067003) and LiFeAs (PRB 94, 201109). Therefore, the observations in this manuscript is not surprising and is not entirely new.

Regarding the Kondo effect interpretation invoked in this manuscript, it is a viewpoint to look at the coherence-incoherence crossover, but it is not required. The coherence-incoherence crossover is an intrinsic property of Hund's metals. One does not need to invoke Kondo effect to explain it.

In summary, this manuscript does not have enough new physics nor significant advance to warrant its publication in Nature Communications.

The manuscript by Younsik et al presents an original work on antiferromagnetic FeTe using combined ARPES, transport, and STM work. They report a Kondo lattice behavior in a d-electron system. Quite remarkably, they find a positive correlation between the AFM order and the Kondo effect.

I have a number of questions which need to be addressed.

1. What is the value of the ordered magnetic moment (say from neutron scattering) and the effective paramagnetic moment? Can these values explain the Kondo effect?
2. From the ARPES measurements, the quasiparticle peak onsets at the magnetic transition temperature. It also looks like a phase transition rather than a crossover. On the other hand, from the STM data at 80K, a broadened Fano lineshape remains. Can the authors address the discrepancy?
3. Why is the hybridization gap seen in the STM data, but not observed in ARPES? Is this a resolution effect?
4. The gamma value of the Fano fit to the STM data is not given in the manuscript. This is an important value that should be compared with transport.
5. How can one rule out the fact that the observed gap in the STM and the quasiparticle peak in ARPES are not due to AFM? Can the authors measure the temperature dependence of the STM spectra to see the onset temperature of the gap? based on their transport measurements, the coherence temperature is 30K. however ARPES does not show any anomaly at that temperature and STM data is missing.
6. What will be the effect of the excess Fe atoms? From the STM topograph one can see a number of excess atoms on the surface. What is the exact composition (y) of the sample used (Fe_{1+y}Te) ? The Fe atoms on the surface can in principle cause the Kondo effect (not the Kondo lattice effect). Can the authors address this?

We would like to thank the reviewers for the useful comments which helped us improve our manuscript. We did our best to answer the questions and comments raised by the reviewers, not only by providing further discussions but also by conducting additional experiments and theoretical calculations which support our claims. Following the suggestions by the reviewers, we added new experimental and theoretical results to the Supplementary Information. A list of major changes in the revised manuscript is given below, followed by point-by-point responses to the reviewers' comments.

List of major changes.

1. Sentences describing the pocket dependence of quasiparticle behaviors are added to the Discussion section, along with newly obtained temperature-dependent ARPES data on the X pocket in the Supplementary Information.
2. DFT and DMFT calculations were additionally conducted to support the Kondo scenario. These results are added to the Supplementary Information. Names of three authors (Dongwook Kim, Minjae Kim, and Jihoon Shim) were added to the author list.
3. The origin of the hump structure seen in ARPES is further discussed.
4. Possibility of different Kondo behaviors at the surface and in the bulk are discussed.
5. A sentence describing the persistent Fano line shape above the magnetic transition temperature is added to the section for Fano line shape and hybridization gap.
6. The value of the Fano line shape width is added to the Result section.
7. The coherent-incoherent temperature is revised from 30 K to 15 K, based on the detailed analyses of the resistivity and heat capacity results. The detailed analysis results are added to the Supplementary Information.
8. Temperature-dependent STS measurements were newly conducted to find the onset temperature of the gap feature. The results are added to the Supplementary Information.
9. Precise resistivity measurement results below 3 K are added to the Supplementary Information for the discussion of possible Kondo scattering induced by excess iron atoms.

Reply to Reviewer 1's report

The manuscript by Younsik Kim et al. presents the results of investigations of FeTe system by means of transport measurements, laser-based ARPES, STM/STS measurements and modelling. The most essential result is the observation of Kondo lattice behavior in FeTe and its cooperation with long-range magnetic order.

In general, I agree with the interpretation of the obtained data. However, I believe that the manuscript, in its actual form, is not complete. I have the following points, which should be discussed and addressed in the revised draft.

Author's response: We are very happy to hear the reviewer's positive opinions on our work. The reviewer's valuable comments, especially on the pocket dependence of transport properties, were very helpful in improving our manuscript. We made every effort to address the points raised by the reviewer.

(1). The authors discuss the spectral pattern and its temperature behavior only in the vicinity of the Gamma point. It is unclear why only this region of BZ was considered. I fully understand that laser-based PE allows probing mainly this region. However, from the PRL 110, 037003 (2013), it looks so that the area around other regions like close to the X-point could be of interest as well. Please, make a comment on that.

Author's response: We agree with the reviewer that the spectral weight near the Brillouin zone corner as well as near the Γ point can affect the transport properties. Before moving on to a detailed discussion, we would like to point out that the spectral weight near the M point (defined in the 2-Fe Brillouin zone scheme) is negligible, while that near the X point shows a rather strong spectral weight¹⁻³. For this reason, we will focus on the X point pocket, which has a much stronger spectral weight near E_F .

To figure out the temperature dependence of the X point pocket spectral weight, we additionally conducted ARPES measurements near the X point region. Figure R1a shows the Fermi surface taken with 21 eV light. As can be seen, the Fermi surface map shows strong spectral weight near the X point as well as the Γ pocket. Temperature dependence of energy distribution curves (EDCs) at the X point is shown in Fig. R1(b). Contrary to the Γ point data, which show a strong temperature dependence and a quasiparticle peak (QP) at low temperature, there is no significant temperature dependence and no signature of quasiparticle peak at the X point. The spectra near the Fermi level follow a usual Fermi-Dirac distribution. It clearly shows that the temperature dependent behavior at the X point is not consistent with the transport properties across T_N . A drastic change in transport properties across the magnetic transition should be accompanied by a change in the Fermi surface and corresponding QP, which is not the case for the X point pocket. Taking these points into account, we would like to point out that Γ point pocket dominates the Kondo behavior observed by transport and STM measurements. To clarify the issue of the pocket dependence in the transport properties, we added Fig. R1 to the Supplementary Information (SI) as Fig. S1 and a sentence in the Discussion section as follows:

Revised (added): "It is also noteworthy that such a strong QP and its strong temperature dependence are only observed near the Γ point (see Supplementary Information for the temperature dependent ARPES results on the X point pocket.)."

Fig. R1. Temperature dependent evolution of the X-point pocket. **a.** Fermi surface map measured at 6 K. **b.** Extracted temperature-dependent energy distribution curves (EDCs) at the red dashed line in **a**. The color bar denotes the temperature of each EDC.

(2). The authors say that "...The Kondo lattice behavior as a result of hybridization between localized Fe 3d_{xy} and itinerant Te 5p_z orbitals." I did not find that this statement is supported neither from experiment nor from theory.

Author's response: We appreciate the reviewer for raising the point. For the Kondo lattice behavior, we believe that our comprehensive experimental results obtained from ARPES, STM, and transport should support our interpretation. As supporting evidence for the Kondo lattice behavior, we showed i) enlargement of the Fermi surface as temperature lowers, ii) Fano line shape and narrow gap feature and iii) T²-resistivity at low temperatures and logarithmic resistivity at high temperatures.

On the other hand, the direct experimental evidence for the orbital character of the heavy band was not provided in the submitted manuscript as the reviewer correctly pointed out. Even though we have experimentally confirmed that the electron band at the Γ point has mainly Te 5p_z orbital character based on the k_z-dependent ARPES data (Fig. 3) and previous first-principles calculation studies^{4,5}, direct evidence for the Fe 3d_{xy} orbital character was not provided.

To investigate the orbital character of the band that hybridizes with the dispersive Te p_z band, we additionally conducted DFT-LDA calculations on paramagnetic FeTe based on experimentally obtained crystal structures⁶. Here, it should be pointed out that DFT cannot directly show many-body effects such as orbital-selective Mott phase and resultant Kondo hybridization. However, it may be utilized as a base layer for predicting Kondo hybridization.

Shown in Fig. R2 are Fe d_{xy} and Te p_z projected DFT calculation results. The thickness of a band represents the projected weight of the corresponding orbital. The band indicated by black arrows corresponds to what we have measured by ARPES. The band is of dominant Te p_z character but also has appreciable Fe d_{xy} character, implying that the two orbitals are strongly hybridized. These results theoretically hints that the experimentally measured dispersive band has Fe d_{xy} orbital character as well as Te p_z orbital character. We note that other bands crossing E_F does not hybridize with p_z band since all other band has even parity whereas p_z band has odd parity⁴.

As an additional supporting argument for the orbital character, we note that only the d_{xy} orbital among t_{2g} orbitals of FeTe is localized, namely orbital-selective Mott phase (OSMP)⁷⁻⁹, whereas other t_{2g} orbitals remain as itinerant^{8,10}. Considering a local magnetic moment is an essential prerequisite for Kondo physics¹¹, the local magnetic moment from the d_{xy} orbital is expected to host Kondo features of FeTe. To provide the aforementioned supporting arguments, we added Fig. R2 to the Supplementary Information and revised the manuscript as follows:

Revised (added):

“The argument is based on the fact that only the d_{xy} orbital of FeTe is in a localized state, which is a prerequisite for the Kondo effect. Density functional theory calculations also confirm the band we measured in ARPES has Te p_z and Fe d_{xy} orbital characters (see the Supplementary Information).”

Fig. R2. Density functional calculation results on FeTe. The obtained electronic structure is projected on Fe d_{xy} (left) and Te p_z (right). The black arrows indicate the band which corresponds to the dispersive band measured by ARPES. $d_{xy}(+)$ and $d_{xy}(-)$ denote even and odd parity d_{xy} band for the inversion symmetry, respectively. The thickness of the bands represents the weight of projected orbital characters.

(3). I should also say that the absence of the results from *ab-initio* calculations is a weak point of the paper, which is considered for publication in NC.

Recently, it was shown that DMFT and DFT calculations are quite useful for unveiling the Kondo lattice properties in *f*-materials:

<https://www.pnas.org/doi/epdf/10.1073/pnas.2001778117>

for prediction and observation of Rashba SOC phenomena in *f*-systems:

<https://onlinelibrary.wiley.com/doi/10.1002/aelm.202100768>

Author’s response: We thank the reviewer for raising the issue. We agree with the reviewer that our claims on Kondo properties can be made more solid with *ab-initio* calculations. Reflecting the review’s comment, we additionally conducted DFT + DMFT calculations on *antiferromagnetic* FeTe. We note that the prediction of Kondo lattice properties in an antiferromagnetic state by means of DMFT calculations is difficult due to residual magnetic moments (the calculations in the references given by the reviewer were for non-magnetic states and had much less convergence issues). Still, we found that the strongly dispersive band along the k_z direction indeed crosses the Fermi level along the Γ -Z line. In addition, we have also identified a signature of hybridization between relevant bands in the

antiferromagnetic state. The details are as follows.

Shown in Fig. R3 is the DMFT calculation result along the Γ -Z line. The magnetic ground state in the DMFT calculation successfully converged to the bicollinear antiferromagnetic state. The strongly dispersive band (p_z) and strongly localized band cross each other near the Fermi level as indicated by the blue dashed circle. This is an indication of the Kondo hybridization, where the localized and itinerant bands cross each other near the Fermi level, even though we could not observe the hybridization gap due to the finite broadening effect of the DMFT calculations. The DMFT calculation result shown in Fig. R3 is added to the Supplementary Information.

Fig. R3. Dynamic mean field theory (DMFT) calculations on FeTe.

(4). I have to say that ARPES data were accurately obtained, treated and presented. This is seen from figure 2c where the correctly applied normalization allows to detect the QP peak, “hump” structure and their temperature evolution.

I suggest to extend the discussion about the “hump” structure. It is still unclear where it comes from and what is its origin.

Author’s response: We thank the reviewer for his/her valuable comment to improve the description of the hump structure. We agree that the explanation of the hump structure in the manuscript is insufficient. We would like to extend the discussion about the hump structure reflecting the reviewer’s comment.

A general feature of Kondo lattice systems is the development of coherency as the temperature is lowered¹². The developed coherency leads the system to a Fermi liquid state at low temperatures, characterized by T^2 -resistivity behavior and a well-defined QP observed in ARPES¹³. On the other hand, the high-temperature state can be understood as an incoherent state due to the strong Kondo scattering. It is well established that a strong scattering results in loss of QP spectral weight near E_F and its transfer to a higher binding energy region^{13,14}. In other words, the hump structure is the incoherent state of the original band, induced by strong Kondo scattering.

Therefore, the temperature-dependent spectral weight at E_F ($E_F - 0.1$ eV) shown in Fig. 2e can be a measure of the coherency (incoherency) of the system, demonstrating the temperature-dependent tendency of the Kondo lattice system. We added the discussion on the origin of the hump structure in the revised manuscript.

Original:

“From the electronic structure point of view, the strong scattering induced in the Kondo scattering regime leads to a loss of coherence which results in a strong hump structure at high binding energy and weak spectral weight near the Fermi level.

Revised:

“From the electronic structure point of view, the strong scattering in the Kondo scattering regime results in breakdown of a well-defined quasiparticle, which in turn leads to loss of spectral weight and its transfer to a higher binding energy region. Thus, the hump structure is the incoherent counterpart of the QP, supported by the spectral weight transfer as shown in Fig. 2e.

(5). Recently, it was shown that the Kondo lattice properties at the surface and in the bulk could be strongly and unexpectedly different:

<https://www.nature.com/articles/s41535-020-00273-7>.

<https://www.nature.com/articles/ncomms11029>

In this respect, what the authors could say concerning FeTe.

Author's response: We thank the reviewer for his/her expert advice on the potential difference between the surface and bulk Kondo lattice properties. The raised point alludes to possible discrepancy between results from bulk- and surface-sensitive measurements. We have utilized transport measurements, ARPES, and STM to observe the Kondo lattice behavior in FeTe. The transport measurement is sensitive to bulk properties of materials, whereas ARPES and STM measure surface properties. Therefore, the quantitative value (i.e. Kondo temperature) extracted from each measurement can be unexpectedly different considering the previous studies¹⁵⁻¹⁷.

The estimated onset temperature of the Kondo scattering of FeTe is around room temperature, as evidenced by insulating resistivity up to room temperature. The Kondo temperature extracted from STM measurements (from the width of the Fano line shape) is 281 K. The two values are similar despite their surface/bulk sensitivities. The surface-bulk difference in Kondo properties in CeRh_2Si_2 or SmB_6 might be attributed to their three-dimensional crystal structure¹⁵⁻¹⁷. Considering FeTe has a quasi-two dimensional crystal structure with a relatively weak interlayer coupling, the surface-bulk difference in Kondo properties may not be significant for FeTe. However, we cannot completely rule out the scenario that the Kondo properties may be different at the surface and in the bulk as suggested in recent studies. To provide more information on this, we added the discussion on the possible surface-bulk difference in Kondo properties and cite the aforementioned papers in the revised manuscript.

Revised (added):

“We note that recent ARPES and STS studies on CeRh_2Si_2 and SmB_6 reported significantly different

Kondo properties at the surface and in the bulk³³⁻³⁵. In such cases, considering the surface sensitivity of ARPES and STS, the Kondo-related properties of FeTe observed via ARPES and STS can be different from those of transport measurements. However, the crystal structure of FeTe is quasi-two-dimensional, which is distinct from CeRh₂Si₂ and SmB₆²⁰. This feature might be the reason for the consistency in the Kondo properties of FeTe observed by ARPES and transport measurements.”

33. Poelchen, G. et al. *npj Quantum Mater.* 5, 70 (2020).

34. Patil, S. et al. *Nat. Commun.* 7, 11029 (2016).

35. Jiao, L. et al. *Nat. commun* 7, 1-6 (2016).

Reply to Reviewer 2's report

1. What is the value of the ordered magnetic moment (say from neutron scattering) and the effective paramagnetic moment? Can these values explain the Kondo effect?

Author's response: We appreciate the reviewer's expert comment on the Kondo effect. A previous neutron scattering study on FeTe reported that the ordered magnetic moment at 10 K was about $2.7 \mu_B$, whereas the effective paramagnetic moment above T_N was about $4 \mu_B$ ¹⁸. Since the Kondo screening reduces the magnetic moment at low temperatures, the experimental values indeed can be explained within the Kondo scenario. We cordially point out that this argument is already discussed in the first paragraph of the Discussion section.

Original:

"The Kondo hybridization scenario is further supported by previous inelastic neutron scattering measurements on FeTe: the study reported that the local magnetic moment of FeTe is $S = 1$ at 10 K but it unexpectedly grows to $S = 3/2$ at 300 K, suggesting low-temperature Kondo screening of the local moments by itinerant electrons²⁶."

2. From the ARPES measurements, the quasiparticle peak onsets at the magnetic transition temperature. It also looks like a phase transition rather than a crossover. On the other hand, from the STM data at 80K, a broadened Fano lineshape remains. Can the authors address the discrepancy?

Author's response: The reviewer raised a question about the persistent Fano line shape at 80 K, which seems inconsistent with the ARPES data. Here is the detailed explanation for the persistent Fano line shape at 80 K. First, we would like to point out that the onset temperature of the Kondo scattering in FeTe is around the room temperature, as evidenced by the insulating resistivity behavior above T_N (Fig. 1c) and the extracted Kondo temperature from the Fano linewidth ($k_B T_K \approx \Gamma$, $T_K = 281$ K) (see Comment 4 for details on the extracted Fano linewidth). Considering the fact that even single-ion Kondo scattering can induce the Fano line shape in the STS spectra^{19,20}, broadened Fano line shape is also expected in the Kondo scattering regime of FeTe where coherence is strongly suppressed.

On the other hand, the quasiparticle peak (QP) intensity can be understood as the degree of coherence developed in the system since the QP is a result of Kondo hybridization and is only well-defined in coherent systems. The emergent QP below T_N indicates that the coherence is suddenly developed in the system upon antiferromagnetic transition. Therefore, the QP in ARPES mainly depends on the coherency of the corresponding system, whereas the Fano line shape is not highly dependent on the coherency. The different requirements for the coherency explain the difference between ARPES and STM data. We clarified this point in the Discussion section of the revised manuscript.

Revised (added):

“The broadened but persistent Fano line shape at 80 K also indicates the system is still in the Kondo scattering regime, while strongly suppressed coherency above T_N leads to the loss of the QP.”

3. Why is the hybridization gap seen in the STM data, but not observed in ARPES? Is this a resolution effect?

Author's response: As the reviewer correctly guessed, the reason why the hybridization gap seen in the STM data is not observed in the ARPES comes from the fact that (1) there is a difference in the thermal broadening of the ARPES and STM data and (2) ARPES sees only the occupied states. ARPES experiments were performed at 15K while STM measurements were done at 4 K. Even though our laser-based ARPES was performed with a resolution of 2 meV, thermal broadening at 15 K results in a full-width half-maximum broadening of the Fermi edge up to 5 meV. Considering the gap size seen with STM is about 7 meV, the gap feature in the ARPES can be smeared out by the broadening effect. More importantly, while ARPES can probe only the occupied side, the center position of the gap is placed slightly above the Fermi level in the unoccupied side, which is naturally expected in metallic Kondo lattice systems. These two factors contribute to why the gap was not seen in the ARPES data.

4. The gamma value of the Fano fit to the STM data is not given in the manuscript. This is an important value that should be compared with transport.

Author's response: We would like to thank the reviewer for the valuable comment. The fitted Γ value is 24.1 meV and the estimated Kondo temperature from the Γ value is around 281 K ($\Gamma = k_B T_K$). The STS data and the Fano fit are plotted in Fig. R4. The obtained Kondo temperature is similar to the onset temperature of the Kondo scattering as discussed above. In short, the estimated Kondo temperature from the Γ value is consistent with the transport result. We added this argument in the revised manuscript and Supplementary Information.

Revised (added):

“The Fano fit shown as blue circles in Fig. 4a gives a Fano line width (Γ value) of 24.1 meV, which corresponds to the Kondo temperature of about 280 K.”

Fig. R4. Fano fit results of the STS data. The orange line indicates the Fano fit.

5. How can one rule out the fact that the observed gap in the STM and the quasiparticle peak in ARPES are not due to AFM? Can the authors measure the temperature dependence of the STM spectra to see the onset temperature of the gap? based on their transport measurements, the coherence temperature is 30K. however ARPES does not show any anomaly at that temperature and STM data is missing.

Author's response: The reviewer raised a valid and important issue. We also agree that we cannot completely rule out the scenario in which the observed gap may be from the AFM. To clarify the issue, we performed additional temperature-dependent STS measurements as shown in Fig. R5. As indicated by the black arrows, the gap feature disappears at 12 K or slightly higher, which is much lower compared to T_N (68 K). From this result, we can rule out the scenario that the gap directly comes from the AFM.

The reviewer also questioned whether the QP observed in ARPES could be due to the AFM. Our temperature-dependent ARPES results show that the Fermi surface and associated electronic structure drastically change upon the magnetic transition. If the changes were indeed induced by AFM order, AFM can change the electronic structure in two ways: (i) exchange splitting^{21,22}, and (ii) zone folding^{23,24}. The exchange splitting scenario, however, cannot explain the mass enhancement of the QP at low temperatures. Our ARPES results revealed that the main orbital character of the QP is Te 5p_z, which necessarily requires external modulation to account for the unexpectedly heavy mass of the p orbital since the exchange splitting can only shift bands in the energy direction. For the zone-folding scenario, the heavy QP could emerge via hybridization between the original and zone-folded bands. Considering the antiferromagnetic ordering vector of FeTe, the X point is overlaid on the Γ point in the AFM Brillouin zone (See Fig. R6). If the QP at the Γ point is a result of the zone-folding, the QP should also be seen at the X point. Our temperature-dependent ARPES results on the X point pocket in Fig. R7 show no signature of QP or strong temperature dependence. From these results, we can rule out that the AFM directly induces the QP seen in ARPES.

Lastly, we would like to discuss the anomaly at the coherent temperature in STS and ARPES data. Before going into details, we would like to make a correction on the coherence temperature, from 15 K to 30 K. The inaccurate initial estimation was made without careful analyses of the transport data. New STS data made us look at the transport data more carefully (in that regard, we thank the reviewer again for suggesting we take more detailed temperature-dependent STS data). Precise analyses of

transport results in Fig. R8 show the coherence temperature to be around 15 K at which the resistivity and heat capacity deviate from Fermi liquid behavior. This coherence temperature roughly coincides with the gap-closing temperature shown in the STM result (12 K). Meanwhile, ARPES measurements were performed at the lowest temperature we can get at that time (15 K). Therefore, it is difficult to see the anomaly with this data. The precisely determined deviation temperature is worth mentioning in the revised manuscript. Hence, we changed the deviation temperature from 30 K to 15 K and added Fig. R5 and R8 in the Supplementary Information.

Original:

“More specifically, it shows a Fermi liquid behavior below 30 K with a T^2 dependence resistivity, and a T-linear behavior between 30 K and 70 K.”

Revised:

“More specifically, it shows a Fermi liquid behavior below 15 K with a T^2 dependence resistivity, and a T-linear behavior between 30 K and 70 K.”

Original:

“ C_v/T deviates from T^2 behavior around 30 K.”

Revised:

“ C_v/T deviates from T^2 behavior around 15 K (see Supplementary Information for the determination of the deviation temperature).”

Fig. R5. Temperature-dependent STS results. Spectra are offset arbitrarily for the sake of a better view. The black dashed line denotes the gap position estimated from the 4 K data and black arrows

indicate the peak position originated from the gap feature.

Fig. R6. Schematic of Brillouin zone of FeTe. PM denotes the paramagnetic state, and AFM denotes the antiferromagnetic state. The red rectangular denotes a reduced Brillouin zone in the AFM state in FeTe.

Fig. R7. Temperature evolution of the X-point pocket. **a.** Fermi surface map measured at 6 K. **b.** Extracted temperature-dependent energy distribution curves (EDCs) at the red dashed line in **a**.

Fig. R8. Coherent-incoherent crossover seen in transport measurements. **a.** Temperature derivative of resistivity. The black dashed line is a guide to the eye. **b.** Temperature-dependent C_v/T .

6. What will be the effect of the excess Fe atoms? From the STM topograph one can see a number of excess atoms on the surface. What is the exact composition (y) of the sample used (Fe_{1+y}Te)? The Fe atoms on the surface can in principle cause the Kondo effect (not the Kondo lattice effect). Can the authors address this?

Author's response: We thank the reviewer for his/her keen observation and valuable comment on the excess Fe issue in FeTe. As the reviewer correctly pointed out, the STM topograph indeed shows a number of excess Fe, while our previous energy dispersive X-ray (EDX) measurements exhibit an almost stoichiometric ratio between Fe and Te. We speculate that this discrepancy stems from the low accuracy of the EDX measurements even with high repeatability²⁵. Thus, we additionally conducted inductively coupled plasma atomic emission spectroscopy measurements to cross-check the ratio of the excess Fe. The results show that the samples have an excess Fe of 0.11. In addition, STM topographic image is also utilized to infer the excess iron content, showing an excess Fe up to 0.12 (counting all the 'bright spots' in the image but some of the bright spots come from subsurface excess iron atoms). These results on excess iron content are summarized in Table R1.

Even though we cannot exactly pinpoint the excess iron content due to the low accuracy of the utilized tools, the relatively high T_N (68 K), and the sharp drop in resistivity upon magnetic transition as shown in Fig. 1 are characteristics of the FeTe samples with relatively low contents of excess Fe^{26,27}, which have been utilized in the previous studies^{1,10,28-30}. We note that T_N of Fe_{1+y}Te ranges from 60 K to 70 K, depending on excess iron content; higher T_N is expected for lower excess iron content^{26,27}. Considering the experimental results shown in Table R1 and previous results on the excess iron content in FeTe, the excess iron content of FeTe utilized in this work should be in the range from 0.08 to 0.12.

Tools	y in Fe_{1+y}Te
EDX	0.00
ICP	0.11
STM	Up to 0.12

Table R1. Excess iron contents in Fe_{1+y}Te measured with various tools.

As the reviewer also remarked, the excess Fe is expected to induce the Kondo effect. More precise resistivity measurements at low temperatures exhibit a resistivity minimum at around 2.2 K as shown in Fig. R9, which can be attributed to the Kondo effect induced by excess Fe. However, we would like to emphasize that this is different from the Kondo lattice behavior at a much higher temperature scale discussed in the manuscript. For example, STS results presented in the main manuscript are conducted away from these excess iron atoms. This implies that the Fano line shape and the narrow gap are intrinsic properties of FeTe.

Along with precisely determined stoichiometry, the effect of excess Fe from the perspective of the Kondo effect is worth mentioning in the revised manuscript. Hence, we added the following discussion to the revised manuscript and Fig. R9 to the Supplementary Information.

Revised (added):

“It is also noteworthy that the resistivity shows a minimum at around 2.2 K (see Supplementary

Information for the corresponding data and discussion).”

Original:

“The stoichiometry was determined by energy dispersive X-ray spectrometry and found to be almost stoichiometric.”

Revised:

“The estimated excess iron concentration of Fe_{1+y}Te is in the range between 0.08 and 0.12, which is determined from STM topography and inductively coupled plasma measurements.”

Fig. R9. Resistivity of FeTe at low temperatures.

Reply to Reviewer 3's report

In this manuscript, the authors reported their angle-resolved photoemission spectroscopy (ARPES), scanning tunneling spectroscopy and transport measurements on FeTe single crystal. They found sharp quasiparticle peak and Fano-type tunneling spectra at low temperatures whereas upon increasing temperature, the spectra weight at the Fermi level decreases and spectra undergoes coherent-incoherent crossover. The authors argued that these observations are manifestations of Kondo lattice behavior which is resulted from the hybridization between the localized Fe 3dxy and itinerant Te 5pz orbitals.

Coherence–incoherence crossover in the normal state of iron-based superconductors was first predicted back in 2009 by Haule and Kotliar (NJP 11, 025021). The coherence-incoherence crossover of FeTe was predicted in 2012 by Yin, Haule and Kotliar (PRB 86, 195141). The temperature evolution of the spectra (ARPES) was shown for FeTe in particular. These and other studies show that Hund's rule coupling plays the dominating role. The coherence-incoherence crossover was observed experimentally in several iron-based superconductors including $AxFe_2-ySe_2$ ($A = K, Rb$) (PRL 110, 067003) and LiFeAs (PRB 94, 201109). Therefore, the observations in this manuscript is not surprising and is not entirely new.

Regarding the Kondo effect interpretation invoked in this manuscript, it is a viewpoint to look at the coherence-incoherence crossover, but it is not required. The coherence-incoherence crossover is an intrinsic property of Hund's metals. One does not need to invoke Kondo effect to explain it.

In summary, this manuscript does not have enough new physics nor significant advance to warrant its publication in Nature Communications.

Author's response: We appreciate the reviewer's professional review of our manuscript, especially from a perspective of coherence-incoherence crossover. While this is an interesting proposal, we respectfully disagree with the reviewer on his/her view of our work.

The main reason for the reviewer's reservation against our work is that the Kondo lattice scenario is not needed to explain the coherence peak below T_N because it can already be explained by the coherence-incoherence crossover seen in other iron-based superconductors. We agree that the appearance of the coherence peak may be explained within Hund's metal scenario. However, one has to note that it can also be explained by the Kondo lattice scenario. Therefore, one has to look further to find out which one of the two is right. We emphasize that the coherence peak is only part of the experimental data we present in this work. We observe evidence for hybridization between two distinct bands along with its strong temperature dependence, leading to a change in the Fermi momentum. The puzzling transport properties of FeTe can be explained by the hybridized band. Temperature-dependent hybridization, the key finding of this work, as well as other experimental observations, cannot be explained within Hund's metal picture. Details are as follows.

The coherence-incoherence crossover in Hund's metal is characterized by a coherent energy scale³¹⁻³³. The crossover can be realized by tuning binding energy³² or temperature³⁴. Our ARPES results on FeTe reveal such a crossover upon changing temperature; a well-defined quasiparticle peak near the Fermi level at low temperatures is broadened upon heating, accompanied by spectral weight transfer to the higher binding energy region. Similar experimental results are already reported in previous studies on other iron-based superconductors as the reviewer pointed out³⁴⁻³⁷. Considering the fact that FeTe is deep in Hund's metal regime⁷⁻⁹, it is seemingly natural to attribute the coherence-incoherence crossover seen in FeTe to the feature of Hund's metals.

However, there are several experimental observations that cannot be addressed by conventional Hund's metal picture. The key finding is that the strong quasiparticle peak is a result of temperature-

dependent hybridization, which accompanies the modulation of the Fermi surface. Even though Kondo physics and Hund physics share similar features such as the coherence-incoherence crossover, the hybridization and resultant Fermi surface modulation is an exclusive behavior in Kondo physics where the itinerant and localized band are hybridized at low temperatures. Thus, we argue that FeTe is distinct from conventional Hund's metals, which necessarily requires further scenarios to address it. Below are detailed experimental observations supporting the Kondo hybridization scenario:

1. Fano line shape and gap feature in scanning tunneling spectroscopy measurements.

Fano line shape in tunneling experiments arises under the presence of two interfering tunneling paths³⁹. The prime example showing the Fano line shape is Kondo systems, where two independent tunneling paths are made through itinerant electrons and heavy quasiparticles^{19,39,40}. Since itinerant electrons and heavy quasiparticles are essential elements of Kondo systems, the Fano line shape is a general and representative feature of Kondo systems.

Our scanning tunneling spectroscopy results on FeTe as shown in Fig. R10 demonstrate the emergence of the Fano line shape at low temperatures. In addition, we would like to address that the narrow gap feature near the Fermi level and its disappearance upon heating are general features of Kondo lattice systems as well, which inevitably requires the Kondo scenario other than the coherence-incoherence crossover picture.

Fig R10. STS results on FeTe. **a.** Differential conductance (dI/dV) spectrum measured on FeTe surface at 4.3 K. The blue circles represent the Fano fitting of the Kondo resonance. The inset shows the position where the spectrum is taken. **b.** dI/dV spectrum enlarged around the Fermi energy. The inset is the spectrum after subtracting the smoothly-varying background. **c.** dI/dV spectra measured at 80 K. **d.** Zoomed-in dI/dV spectrum.

2. Modulation of the Fermi surface as a function of temperature.

A well-known feature of Kondo lattice systems is the enlargement of the Fermi surface. By creating heavy quasiparticles via Kondo interaction at low temperatures, localized electrons start to participate in conduction, resulting in the enlargement of the Fermi surface and a decrease of the Fermi velocity¹¹. The two features are indeed observed in our study (Figs. R11a and R11b). On the other hand, other iron-based superconductors showing coherence-incoherence crossover do not show such features. In particular, a previous ARPES study on LiFeAs reported precise temperature-dependent ARPES measurement, but the results do not show any signature of change in the Fermi momentum and Fermi velocity upon changing temperature³⁴.

Fig. R11. Temperature-dependent Fermi momentum and Fermi velocity of FeTe. (a and b) Temperature-dependent Fermi momentum (k_F) and Fermi velocity (v_F), respectively, obtained from ARPES. Errors bars in (a and b) represent the fitting errors of Fermi momentum and Fermi velocity, respectively.

3. Emergent electron band at the Γ point originated from Te p orbitals.

Our laser- and synchrotron-based ARPES measurements show an unexpected electron band and corresponding strong quasiparticle peak at the Γ point of FeTe, which is not observed in other iron-based superconductors. A previous study on the coherence-incoherence crossover in FeTe based on DFT + DMFT failed to predict the electron band as well⁴¹. We further figured out that the electron band originates from Te $5p_z$ orbitals, and the Te band exclusively determines the metallic transport properties at the antiferromagnetic state in FeTe.

The Hund physics and related coherence-incoherence crossover in iron-based superconductors have been mainly and exclusively focused on Fe d-orbitals⁹. Even though the coherence-incoherence crossover is well established in Fe(Te, Se) compounds based on the Hund physics, the origin of the metallic antiferromagnetic state at the Te end was still controversial^{1,3,10}. Our experimental results attribute the metallic behavior to Te p orbitals, which cannot be accounted for by conventional Hund physics.

Considering the above three points that we addressed, we believe that our experimental observations clearly demonstrate that FeTe cannot be fully described by the conventional coherence-incoherence crossover in Hund physics. Indeed, recent theoretical work proposed that the interorbital hopping in the orbital-selective Mott phase can develop a narrow quasiparticle peak near the Fermi level³⁸. In this perspective, our work emphasizes the role of interorbital coupling rather than intraorbital coupling, which was the main focus of conventional Hund physics. The metallic behavior originates from p orbitals and the emergent Fano line shape are novel features of FeTe, which makes FeTe distinct from other iron-based superconductors.

Even though our observations can be reconciled by the Kondo lattice picture, what the reviewer proposed is worth being mentioned. Therefore, we added a sentence to the discussion along with relevant references including the two theoretical proposals the reviewer mentioned.

Revised (added):

Here, it is also noteworthy that the appearance of the coherence peak may be accounted for within the coherence-incoherence crossover picture in Hund's metal^{26,27} as observed in some of the iron-based superconductors²⁸⁻³⁰. However, the Kondo hybridization picture is needed to explain the other aspects of the experimental results. Indeed, recent theoretical work proposed that the interorbital hopping in the orbital-selective Mott phase can develop a narrow quasiparticle peak near the Fermi level³¹. In this perspective, our work emphasizes the role of interorbital coupling.

26. Haule, K., and Kotliar, G., *New J. Phys.* **11**, 025021 (2009).
27. Yin, Z. P., Haule, K., and Kotliar, G., *Phys. Rev. B* **86**, 195141 (2012).
28. Yi, M. et al. *Nat. Commun.* **6**, 7777 (2015).
29. Yi, M. et al. *Phys. Rev. Lett.* **110**, 067003 (2013).
30. Miao, H. et al. *Phys. Rev. B* **94**, 201109(R) (2016).
31. Kugler, F. B., and Kotliar, G., *Phys. Rev. Lett.* **129** 096403 (2022).

References

- 1 Jiang, J. *et al.* Distinct in-plane resistivity anisotropy in a detwinned FeTe single crystal: Evidence for a Hund's metal. *Physical Review B* **88**, doi:10.1103/PhysRevB.88.115130 (2013).
- 2 Liu, Z. K. *et al.* Experimental observation of incoherent-coherent crossover and orbital-dependent band renormalization in iron chalcogenide superconductors. *Physical Review B* **92**, doi:10.1103/PhysRevB.92.235138 (2015).
- 3 Liu, Z. K. *et al.* Measurement of coherent polarons in the strongly coupled antiferromagnetically ordered iron-chalcogenide Fe_{1.02}Te using angle-resolved photoemission spectroscopy. *Phys Rev Lett* **110**, 037003, doi:10.1103/PhysRevLett.110.037003 (2013).
- 4 Wang, Z. *et al.* Topological nature of the FeSe_{0.5}Te_{0.5} superconductor. *Physical Review B* **92**, doi:10.1103/PhysRevB.92.115119 (2015).
- 5 Peng, X. L. *et al.* Observation of topological transition in high-T_c superconducting monolayer FeTe_{1-x}Sex films on SrTiO₃(001). *Physical Review B* **100**, doi:10.1103/PhysRevB.100.155134 (2019).
- 6 Mizuguchi, Y., Tomioka, F., Tsuda, S., Yamaguchi, T. & Takano, Y. FeTe as a candidate material for new iron-based superconductor. *Physica C: Superconductivity* **469**, 1027-1029 (2009).
- 7 Yi, M., Zhang, Y., Shen, Z.-X. & Lu, D. Role of the orbital degree of freedom in iron-based superconductors. *npj Quantum Materials* **2**, doi:10.1038/s41535-017-0059-y (2017).
- 8 Huang, J. *et al.* Correlation-driven electronic reconstruction in FeTe_{1-x}Sex. *Communications Physics* **5**, 1-9 (2022).
- 9 Yin, Z. P., Haule, K. & Kotliar, G. Kinetic frustration and the nature of the magnetic and paramagnetic states in iron pnictides and iron chalcogenides. *Nat Mater* **10**, 932-935, doi:10.1038/nmat3120 (2011).

- 10 Lin, P. H. *et al.* Nature of the bad metallic behavior of Fe_{1.06}Te inferred from its evolution in
the magnetic state. *Phys Rev Lett* **111**, 217002, doi:10.1103/PhysRevLett.111.217002 (2013).
- 11 Stewart, S. G. Heavy-fermion systems. *Reviews of Modern Physics* **56**, 755 (1984).
- 12 Coleman, P. Heavy fermions: Electrons at the edge of magnetism. *arXiv preprint cond-
mat/0612006* (2006).
- 13 Jang, S. *et al.* Evolution of the Kondo lattice electronic structure above the transport
coherence temperature. *Proceedings of the National Academy of Sciences* **117**, 23467-23476
(2020).
- 14 Haule, K., Rosch, A., Kroha, J. & Wölfle, P. Pseudogaps in an incoherent metal. *Physical review
letters* **89**, 236402 (2002).
- 15 Patil, S. *et al.* ARPES view on surface and bulk hybridization phenomena in the
antiferromagnetic Kondo lattice CeRh₂Si₂. *Nat Commun* **7**, 11029,
doi:10.1038/ncomms11029 (2016).
- 16 Poelchen, G. *et al.* Unexpected differences between surface and bulk spectroscopic and
implied Kondo properties of heavy fermion CeRh₂Si₂. *npj Quantum Materials* **5**, 1-7 (2020).
- 17 Jiao, L. *et al.* Additional energy scale in SmB₆ at low-temperature. *Nature communications*
7, 1-6 (2016).
- 18 Zaliznyak, I. A. *et al.* Unconventional temperature enhanced magnetism in Fe_{1.1}Te. *Physical
review letters* **107**, 216403 (2011).
- 19 Wahl, P., Seitsonen, A., Diekhöner, L., Schneider, M. A. & Kern, K. Kondo-effect of
substitutional cobalt impurities at copper surfaces. *New Journal of Physics* **11**, 113015 (2009).
- 20 Uchihashi, T., Zhang, J., Kröger, J. & Berndt, R. Quantum modulation of the Kondo resonance
of Co adatoms on Cu/Co/Cu (100): Low-temperature scanning tunneling spectroscopy study.
Physical Review B **78**, 033402 (2008).
- 21 Vidal, R. *et al.* Surface states and Rashba-type spin polarization in antiferromagnetic MnBi
₂Te₄ (0001). *Physical Review B* **100**, 121104 (2019).
- 22 Schruck, B. *et al.* Emergence of Fermi arcs due to magnetic splitting in an antiferromagnet.
Nature **603**, 610-615 (2022).
- 23 Watson, M. D. *et al.* Three-dimensional electronic structure of the nematic and
antiferromagnetic phases of NaFeAs from detwinned angle-resolved photoemission
spectroscopy. *Physical Review B* **97**, 035134 (2018).
- 24 Song, D. *et al.* Electron number-based phase diagram of Pr_{1-x}La_xCuO_{4-δ} and possible
absence of disparity between electron-and hole-doped cuprate phase diagrams. *Physical
review letters* **118**, 137001 (2017).
- 25 Carlton, R. A., Lyman, C. E. & Roberts, J. E. Accuracy and precision of quantitative energy-
dispersive x-ray spectrometry in the environmental scanning electron microscope. *Scanning:
The Journal of Scanning Microscopies* **26**, 167-174 (2004).
- 26 Machida, T. *et al.* Effect of excess Fe on magnetic properties and crystallographic phases in
Fe_{1+δ}Te. *Physica C: Superconductivity* **484**, 19-21 (2013).

- 27 Liu, L. *et al.* Reversed anisotropy of the in-plane resistivity in the antiferromagnetic phase of iron tellurides. *Physical Review B* **91**, 134502 (2015).
- 28 Hanke, T. *et al.* Reorientation of the diagonal double-stripe spin structure at Fe_{1+y}Te bulk and thin-film surfaces. *Nat Commun* **8**, 13939, doi:10.1038/ncomms13939 (2017).
- 29 Enayat, M. *et al.* Real-space imaging of the atomic-scale magnetic structure of Fe_{1+y}Te. *Science* **345**, 653-656 (2014).
- 30 Zhang, Y. *et al.* Strong correlations and spin-density-wave phase induced by a massive spectral weight redistribution in α -Fe_{1.06}Te. *Physical Review B* **82**, doi:10.1103/PhysRevB.82.165113 (2010).
- 31 Haule, K. & Kotliar, G. Coherence–incoherence crossover in the normal state of iron oxypnictides and importance of Hund's rule coupling. *New journal of physics* **11**, 025021 (2009).
- 32 Jang, B. G. *et al.* Direct observation of kink evolution due to Hund's coupling on approach to metal-insulator transition in NiS_{2-x}Sex. *Nature communications* **12**, 1-7 (2021).
- 33 Georges, A., Medici, L. d. & Mravlje, J. Strong correlations from Hund's coupling. *Annu. Rev. Condens. Matter Phys.* **4**, 137-178 (2013).
- 34 Miao, H. *et al.* Orbital-differentiated coherence-incoherence crossover identified by photoemission spectroscopy in LiFeAs. *Physical Review B* **94**, 201109 (2016).
- 35 Yi, M. *et al.* Observation of temperature-induced crossover to an orbital-selective Mott phase in A x Fe_{2-y}Se₂ (A= K, Rb) superconductors. *Physical review letters* **110**, 067003 (2013).
- 36 Wu, Y. P. *et al.* Emergent Kondo Lattice Behavior in Iron-Based Superconductors AFe₂As₂ (A=K, Rb, Cs). *Phys Rev Lett* **116**, 147001, doi:10.1103/PhysRevLett.116.147001 (2016).
- 37 Miao, H. *et al.* Hund's superconductor Li(Fe,Co)As. *Physical Review B* **103**, doi:10.1103/PhysRevB.103.054503 (2021).
- 38 Kugler, F. B. & Kotliar, G. Is the orbital-selective Mott phase stable against interorbital hopping? *Physical review letters* **129**, 096403 (2022).
- 39 Újsághy, O., Kroha, J., Szunyogh, L. & Zawadowski, A. Theory of the Fano resonance in the STM tunneling density of states due to a single Kondo impurity. *Physical review letters* **85**, 2557 (2000).
- 40 Knorr, N., Schneider, M. A., Diekhöner, L., Wahl, P. & Kern, K. Kondo effect of single Co adatoms on Cu surfaces. *Physical review letters* **88**, 096804 (2002).
- 41 Yin, Z., Haule, K. & Kotliar, G. Fractional power-law behavior and its origin in iron-chalcogenide and ruthenate superconductors: Insights from first-principles calculations. *Physical Review B* **86**, 195141 (2012).

REVIEWER COMMENTS

Reviewer #1 (Remarks to the Author):

I am satisfied with the answers given to my comments by the authors of the above mentioned paper. The authors have changed the text of their manuscript accordingly. In my opinion, the paper is now acceptable for publication.

Reviewer #2 (Remarks to the Author):

The authors have fully and adequately responded to all my raised questions and comments. The revised manuscript is now more complete and provides relevant information on the interplay of Kondo lattice effect and magnetism in FeTe. I do not have any further comments and recommend that the manuscript be published in Nature Communications.

Reviewer #3 (Remarks to the Author):

In the authors' response to my previous report, the authors acknowledge that their ARPES results can be naturally explained by the coherence-incoherence crossover in Hund's metals. The authors argue, however, several experimental observations cannot be explained by the coherence-incoherence crossover, including the Fano line shape and gap feature in the STS measurement, change of Fermi surface as a function of temperature, emergent electron band at Gamma point originated from Te p orbitals. Here are my comments and questions on these points.

1. The emergent electron band at the Gamma point originated from Te p orbitals.

What are the experimental evidences that the electron band at the Gamma point (Fig 2b) originates from Te p orbitals? I don't see any proof in the manuscript. It is purely the authors' guess. However, this guess is clearly incorrect. If this electron band at Gamma point is of Te p orbital character, it should persist to high temperature as the Te p orbital does not undergo a coherence-incoherence crossover in this temperature range. In strong contrast, this electron band cannot be seen at 80 K in their experiment, suggesting it cannot be Te p band. On the other hand, both DFT and DFT+DMFT calculations in the paramagnetic/nonmagnetic state have only hole bands around the Gamma point,

this is consistent with their ARPES at 80 K ($> T_N$). Both DFT and DFT+DMFT calculations in the paramagnetic/nonmagnetic state also have electron bands around the BZ corner (i.e., M point), which are mainly of Fe 3d orbital character. Since the Neel temperature is about 70 K, and the electron band around Gamma point appears only at temperatures below 70 K, it is highly possible that this electron band is a folded electron band from the BZ corner mentioned above. This is because the antiferromagnetic phase transition at 70 K makes the unit cell larger and the BZ smaller, which leads to the band folding from the BZ boundary to the BZ center of the bands in the paramagnetic state.

2. Modulation of the Fermi surface as a function of temperature.

As discussed above, it is unlikely that there is an electron-like Te p band that hybridizes with the Fe 3d xy band along the Gamma-X direction. It is unlikely that the change of the Fermi surface as a function of temperature is due to the Kondo physics. Instead, it may be due to the combined effects of the folding of the bands caused by the antiferromagnetic transition and the coherence-incoherence crossover.

3. Fano line shape and gap feature in scanning tunneling spectroscopy measurements.

First of all, there is no real gap in their raw data in the STS measurement. The gap-like feature is obtained by subtracting a large “background”. The authors did not see a gap-like feature in the ARPES measurement neither. More importantly, as pointed out above, there is no evidence that there is an electron-like Te p band that hybridizes with the Fe 3d xy band along the Gamma-X direction. Therefore, the Kondo picture does not apply here. Lastly, the Fano line shape could be explained by the combined effects of the folding of the bands caused by the antiferromagnetic transition and the coherence-incoherence crossover.

In summary, I am still not convinced by the authors. I don't think the authors' claim of Kondo physics in FeTe is supported by the results presented in the manuscript. Therefore, I don't think the manuscript is suitable for publication in Nature Communications in its present form.

Reviewer #3 (Remarks to the Author):

1. The emergent electron band at the Gamma point originated from Te p orbitals.

What are the experimental evidences that the electron band at the Gamma point (Fig 2b) originates from Te p orbitals? I don't see any proof in the manuscript. It is purely the authors' guess. However, this guess is clearly incorrect. If this electron band at Gamma point is of Te p orbital character, it should persist to high temperature as the Te p orbital does not undergo a coherence-incoherence crossover in this temperature range. In strong contrast, this electron band cannot be seen at 80 K in their experiment, suggesting it cannot be Te p band. On the other hand, both DFT and DFT+DMFT calculations in the paramagnetic/nonmagnetic state have only hole bands around the Gamma point, this is consistent with their ARPES at 80 K ($> T_N$). Both DFT and DFT+DMFT calculations in the paramagnetic/nonmagnetic state also have electron bands around the BZ corner (i.e., M point), which are mainly of Fe 3d orbital character. Since the Neel temperature is about 70 K, and the electron band around Gamma point appears only at temperatures below 70 K, it is highly possible that this electron band is a folded electron band from the BZ corner mentioned above. This is because the antiferromagnetic phase transition at 70 K makes the unit cell larger and the BZ smaller, which leads to the band folding from the BZ boundary to the BZ center of the bands in the paramagnetic state.

Authors' response: As far as we understand, the reviewer's objection to our interpretation is based on two accounts; (1) assignment of p orbital character to the electron band at the Gamma point is only a guess without experimental evidences, and (2) the electron band may be accounted for by a folding-induced band scenario. Even though we already presented some evidences on the origin and orbital character of the electron band in previous versions of the manuscript and reply, they may have not been convincing for the reviewer. Therefore, we provide *direct experimental evidences* along with more detailed explanations. More specifically, we conducted new polarization-dependent ARPES measurements that directly show the p orbital character of the band. As for the second point, we will show that the folding scenarios cannot account for the measured data. Detailed point-by-point responses to the comments are provided as follows.

(1) Orbital character of the electron band

Here, we provide three pieces of experimental evidence to support the argument that the electron band has a p orbital character: i) polarization-dependent ARPES intensity, ii) photon energy-dependent spectral weight, and iii) strong k_z dispersion. We believe that the experimental results firmly prove the p orbital character of the electron band. Detailed explanations are provided below.

i) Polarization- and experimental geometry-dependent ARPES measurements

Summarized in Table R1 are the photoemission selection rules of t_{2g} and p_z orbitals near the Γ point as a function of the sample geometry and light polarization^{1,2}. Here, Geometry 1(2) represents the geometry in which the ΓX (ΓM) line of the sample is aligned along the incident photon (see the schematic of the sample geometry on the left side of Fig. R1). Since the orbital character of a band can be determined based on the selection rules presented in Table R1, we performed polarization- and geometry-dependent ARPES to experimentally determine the orbital character.

Figure R1 shows that a circular electron pocket is only visible with p -polarized light, regardless of the sample geometry. Based on the selection rules given in Table R1, such experimental polarization dependence is possible only for the p_z orbital. For example, a d_{xz} orbital band should be visible with s -polarization in Geometry 2, which is opposite to the observation. Hence, we can conclude that the electron band has p_z orbital character. We added these results and associated discussions to the Supplementary Information.

Revised (added)

“Polarization- and experimental geometry-dependent ARPES measurements also confirm the p_z character of the band (see Supplementary Information).”

Geometry 1	p -pol	s -pol	Geometry 2	p -pol	s -pol
d_{xz}	Δ	Δ	d_{xz}	O	X
d_{yz}	Δ	Δ	d_{yz}	X	O
d_{xy}	O	X	d_{xy}	X	O
p_z	O	X	p_z	O	X

Table R1. Sample geometry and polarization dependent photoemission selection rules. Geometry 1(2) denotes the geometry in which the ΓX (ΓM) line of the sample is aligned along the incident photon. Here, the orbitals are defined in the sample frame.

Fig. R1. Geometry- and polarization-dependent Fermi surfaces of FeTe. Γ M(Γ X) denotes the geometry in which the Γ M(Γ X) line of the sample is aligned along the incident photon. The ARPES data are taken with 29 eV light. The X and Y directions are defined in the laboratory frame.

ii) Photon energy-dependent spectral weight

Another experimental supporting evidence for the p orbital character comes from the photon energy-dependent spectral weight. A previous study has reported that the photoionization cross section of Te $5p$ orbitals becomes stronger at low photon energies (i.e. $h\nu < 20$ eV)¹ and far exceeds that of Fe $3d$ orbitals. Indeed, the electron band is strong in our ARPES data taken with 11 eV light, whereas it is not clearly visible with higher photon energies². Therefore, the electron band clearly seen with 11 eV light is another evidence that this band has Te $5p$ orbital character.

iii) Strong k_z dispersion

ARPES measurements along the k_z direction (photon energy dependent ARPES result in Fig. 3 of the manuscript) show that the electron band strongly disperses along the k_z direction. Considering the fact that the Fe $3d$ orbitals are less dispersive along the k_z direction due to the strong correlation^{3,4}, the large dispersion of the electron band along the k_z direction implies that this band has a Te p orbital character. Our DFT calculation results consistently show that a strongly dispersive band near the Fermi level has a significant p orbital character (see Fig. R2).

Fig. R2. Density functional theory calculation results on FeTe. The obtained electronic structure is projected on Fe d_{xy} (left) and Te p_z (right). The black arrows indicate the band which corresponds to the dispersive band measured by ARPES. $d_{xy}(+)$ and $d_{xy}(-)$ denote even and odd parity d_{xy} band for the inversion symmetry, respectively. The thickness of the bands represents the weight of projected orbital characters.

Considering the experimental observations and DFT calculations, we concluded that the electron band has significant p orbital character. In addition, we would like to remark that even though the electron band has significant p orbital character, the band is an antibonding state of p_z and d_{xy} orbital based on the DFT calculations. Thus, some part of the correlation from d -orbitals can be involved. We believe that it explains the unexpectedly low coherence temperature of the p orbital band in FeTe.

(2) Folding-induced electron band scenario

The reviewer also argues that the observed electron band could be a folded band of an electron band at the M point. We respectfully disagree with the folding band scenario for the following reasons. First, we would like to point out that the folding induced by the antiferromagnetism (AFM) in FeTe results in an overlap of Γ and X points, not the M point mentioned by the reviewer, due to the ordering vector of the AFM in FeTe (see Fig. R3)^{5,6}. Hence, if the electron band were due to the folding induced by the AFM, the band would come from the X point. However, the electron band near the Γ point cannot be a result of folding from the X point because there is no Fermi surface pocket near the X point as shown in previous DFT calculation results^{7,8}. In addition, temperature-dependent ARPES results near the X point do not exhibit a quasiparticle as shown in Fig. R4, unlike the temperature-dependent behavior near the Γ point.

Fig. R3. Schematic of the FeTe Brillouin zone. PM denotes the paramagnetic state, and AFM denotes the antiferromagnetic state. The red rectangular denotes a reduced Brillouin zone in the AFM state in FeTe.

Fig. R4. Temperature-dependent evolution of the X-point pocket. **a.** Fermi surface map measured at 6 K. **b.** Extracted temperature-dependent energy distribution curves (EDCs) at the red dashed line in **a.** The color bar denotes the temperature of each EDC.

2. Modulation of the Fermi surface as a function of temperature.

As discussed above, it is unlikely that there is an electron-like Te p band that hybridizes with the Fe 3d xy band along the Gamma-X direction. It is unlikely that the change of the Fermi surface as a function of temperature is due to the Kondo physics. Instead, it may be due to the combined effects of the folding of the bands caused by the antiferromagnetic transition and the coherence-incoherence crossover.

Authors' response: The reviewer argues that the combined effect of folding and coherence-incoherence crossover can induce the modulation of the Fermi surface. However, as we convincingly showed above *with experimental evidences*, the electron band mostly originates from Te p orbitals, not from a band folding. Moreover, the coherence-incoherence itself cannot account for the modulation of the Fermi surface⁹. For these reasons, we respectfully disagree with the reviewer.

On the other hand, the Kondo scenario naturally explains the Fermi surface modulation, from the

perspective of hybridization between two distinct bands. The enlargement of the Fermi surface and decrease in the Fermi velocity can be explained by the hybridization between the itinerant band and the localized band.

3. Fano line shape and gap feature in scanning tunneling spectroscopy measurements.

First of all, there is no real gap in their raw data in the STS measurement. The gap-like feature is obtained by subtracting a large “background”. The authors did not see a gap-like feature in the ARPES measurement neither. More importantly, as pointed out above, there is no evidence that there is an electron-like Te p band that hybridizes with the Fe 3d xy band along the Gamma-X direction. Therefore, the Kondo picture does not apply here. Lastly, the Fano line shape could be explained by the combined effects of the folding of the bands caused by the antiferromagnetic transition and the coherence-incoherence crossover.

The reviewer dismisses our claim of gap observation based on the fact that our gap-like feature (reduction of the local density of state) is sitting on a large ‘background’, asserting that there is no real gap. However, we would like to point out that such background is natural for the following reasons. First of all, the reviewer missed the fact that there are other Fe 3d orbitals, such as d_{xz} and d_{yz} , which do not have gaps. As previous ARPES studies on FeTe show⁵, d_{xz} and d_{yz} bands are not expected to have gaps. These two bands should thus contribute to the finite background. Second, even without such contributing bands, gap features in STS results are often smeared by backgrounds, especially for small gaps due to the instrumental resolution and finite quasiparticle lifetime, as previous studies show¹⁰⁻¹². In fact, a gap with a substantial background is observed in other Kondo systems^{10,11,13}. Considering the facts, the background seen in the gap feature of FeTe is not inconsistent with our interpretation.

The reviewer also pointed out that the gap-like feature is not seen in the ARPES measurements (we would like to remind the reviewer that this comment was already raised by another reviewer in the first round of the review). The main reason stems from the fact that the center position of the gap is placed slightly above the Fermi level (as the STS data shows) and ARPES sees only the occupied states. In the case that an ARPES gap is present, the Fermi level is placed in the gap. This would lead to an increase in the resistivity at low temperatures (namely Kondo insulator), which is not the case for FeTe. On the other hand, the Fermi level of metallic Kondo lattice systems crosses the quasiparticle peak induced by hybridization, which is slightly off the gap¹⁴. Considering the position of the gap and metallic behavior of FeTe at low temperatures, the gap feature should not be seen in ARPES.

The absence of the gap can also be attributed to the difference in measurement temperatures between ARPES and STM. Specifically, while the STM measurement was conducted at 4 K, the ARPES measurement was performed at a higher temperature of 15 K. The increased thermal broadening at a higher temperature could have the gap feature to be smeared out, even if it existed near the Fermi level.

The reviewer insists that the Fano line shape can be explained by the combined effect of folding of the bands and coherence-incoherence crossover. However, the reviewer is unclear on how the Fano line shape can be constructed through the combined effect of folding of the bands. More importantly, as we already pointed out above, the reviewer's folding band scenario cannot explain the electron band below T_N because there is no electron band at the X point in the first place. Another important aspect that rules out the folding interpretation is that the Fano line shape is weakened but persistent above the magnetic transition temperature where resistivity shows insulating behavior. If the Fano line shape is associated with folding, it should disappear upon magnetic transition. The persistent Fano line shape in the insulating regime indicates that the incoherent Kondo interaction (namely Kondo scattering) dominates low-energy electronic correlation¹⁵. Considering the aforementioned discussions, we believe the Kondo picture is more natural to explain the Fano line shape observed in FeTe.

In summary, I am still not convinced by the authors. I don't think the authors' claim of Kondo physics in FeTe is supported by the results presented in the manuscript. Therefore, I don't think the manuscript is suitable for publication in Nature Communications in its present form.

We fully answered all the questions raised by the reviewer with experimental evidences. I hope that the reviewer is now convinced that the observed features are a result of Kondo lattice behavior in FeTe.

References

- 1 Zhang, P. *et al.* Observation of topological superconductivity on the surface of an iron-based superconductor. *Science* **360**, 182-186, doi:10.1126/science.aan4596 (2018).
- 2 Sobota, J. A., He, Y. & Shen, Z.-X. Angle-resolved photoemission studies of quantum materials. *Reviews of Modern Physics* **93**, 025006 (2021).
- 3 Yin, Z. P., Haule, K. & Kotliar, G. Kinetic frustration and the nature of the magnetic and paramagnetic states in iron pnictides and iron chalcogenides. *Nat Mater* **10**, 932-935, doi:10.1038/nmat3120 (2011).
- 4 Huang, J. *et al.* Correlation-driven electronic reconstruction in FeTe_{1-x}Sex. *Communications Physics* **5**, 1-9 (2022).
- 5 Lin, P. H. *et al.* Nature of the bad metallic behavior of Fe_{1.06}Te inferred from its evolution in the magnetic state. *Phys Rev Lett* **111**, 217002, doi:10.1103/PhysRevLett.111.217002 (2013).
- 6 Jiang, J. *et al.* Distinct in-plane resistivity anisotropy in a detwinned FeTe single crystal: Evidence for a Hund's metal. *Physical Review B* **88**, doi:10.1103/PhysRevB.88.115130 (2013).
- 7 Yin, Z., Haule, K. & Kotliar, G. Fractional power-law behavior and its origin in iron-chalcogenide and ruthenate superconductors: Insights from first-principles calculations. *Physical Review B* **86**, 195141 (2012).
- 8 Wang, Z. *et al.* Topological nature of the FeSe_{0.5}Te_{0.5} superconductor. *Physical Review B* **92**, doi:10.1103/PhysRevB.92.115119 (2015).
- 9 Miao, H. *et al.* Orbital-differentiated coherence-incoherence crossover identified by photoemission spectroscopy in LiFeAs. *Physical Review B* **94**, 201109 (2016).
- 10 Jiao, L. *et al.* Additional energy scale in SmB₆ at low-temperature. *Nature communications* **7**, 1-6 (2016).
- 11 Röblier, S. *et al.* Hybridization gap and Fano resonance in SmB₆. *Proceedings of the National Academy of Sciences* **111**, 4798-4802 (2014).
- 12 Giannakis, I. *et al.* Orbital-selective Kondo lattice and enigmatic f electrons emerging from inside the antiferromagnetic phase of a heavy fermion. *Science advances* **5**, eaaw9061 (2019).
- 13 Zhang, Y. *et al.* Emergence of Kondo lattice behavior in a van der Waals itinerant ferromagnet, Fe₃GeTe₂. *Science advances* **4**, eaao6791 (2018).
- 14 Jang, S. *et al.* Evolution of the Kondo lattice electronic structure above the transport coherence temperature. *Proceedings of the National Academy of Sciences* **117**, 23467-23476 (2020).
- 15 Knorr, N., Schneider, M. A., Diekhöner, L., Wahl, P. & Kern, K. Kondo effect of single Co adatoms on Cu surfaces. *Physical review letters* **88**, 096804 (2002).

REVIEWERS' COMMENTS

Reviewer #3 (Remarks to the Author):

In the 2nd revised manuscript and authors' response, the authors provided more experimental results to prove that the electron band near Gamma is Te 5p_z band. However, there are still several issues as following:

1. Table R1. The authors wrote "For example, a dx_z orbital band should be visible with s-polarization in Geometry 2, which is opposite to the observation." However, according to Table R1, d_{xz} orbital band should NOT be visible with s-polarization in Geometry 2, which would be consistent with the observation". The authors should clarify this inconsistency. In addition, the authors should write clearly what the triangle symbol means.
2. According to Figure 2, the electron band at Gamma has a very low coherence temperature. If, as the authors argued, this electron band is Te 5p_z band, why does Te 5p_z band have such low coherence temperature? The argument that it has some small mixture of Fe 3d_{xy} orbital is not convincing.
3. The hybridization gap. The authors argued that they observed a hybridization gap between a Fe 3d_{xy} band and the Te 5p_z band along Gamma-Z path as illustrated in Figure 5. However, according to symmetry and DFT calculations, the crossing of the Fe 3d_{xy} band and Te 5p_z band is protected by symmetry, i.e., there should be no hybridization gap opened between these two bands around the crossing point. The authors should give good reasons why these two bands hybridize and open a gap (what breaks the symmetry protection?)
4. If the electron band at Gamma is Te 5p_z band, it means this Te 5p_z band is below the Fermi level at Gamma point and the entire Te 5p_z band is below the Fermi level along the Gamma-Z path, which contradicts with Figure 5 a.b. and Figure. S9. Therefore, there is no crossing between the Fe 3d_{xy} band and Te 5p_z band along the Gamma-Z path. As a result, no hybridization gap can open between the Fe 3d_{xy} band and Te 5p_z band along the Gamma-Z path. This contradicts with the claim of the manuscript.

Regarding band folding, while the bicollinear antiferromagnetic state (BAFM) is the ground state of bulk FeTe, the stripe ($\pi,0$) antiferromagnetic state (SAFM) is very close to the BAFM state in energy, i.e., the SAFM state competes strongly with the BAFM state. Due to material defects and finite temperature effect, I believe there are band folding from the M point to the Gamma point in the ARPES experiments due to partial SAFM order in the FeTe sample.

In summary, there are still contradicting issues with the revised manuscript as listed above. The manuscript does not have a coherent story. Therefore, I cannot recommend to publish it in Nature Communications.

Reviewer #3 (Remarks to the Author):

In the 2nd revised manuscript and authors' response, the authors provided more experimental results to prove that the electron band near Gamma is Te 5p_z band. However, there are still several issues as following:

1. Table R1. The authors wrote "For example, a dxz orbital band should be visible with s-polarization in Geometry 2, which is opposite to the observation." However, according to Table R1, d_{xz} orbital band should NOT be visible with s-polarization in Geometry 2, which would be consistent with the observation". The authors should clarify this inconsistency. In addition, the authors should write clearly what the triangle symbol means.

The reviewer pointed out the discrepancy between the experimental results and the description associated with Table R1. As the reviewer correctly mentioned, the d_{xz} orbital should NOT be visible with s-polarization in Geometry 2. We made a mistake by omitting NOT in the sentence. The experimental observation shows a circular Fermi surface with s-polarization in Geometry 2. These results reveal that the electron band mainly consists of p_z orbital rather than d_{xz} orbital.

As for the triangle symbols, we meant to indicate a relatively weak transition in comparison to other allowed transitions. However, we realized that a 'weak transition' may not be clearly defined. Since it is still an allowed transition, we decided to we simplify the table by replacing the triangular symbols with circular symbols (this is what is done conventionally). Supplementary Table 1 now shows whether the transition is allowed (O) or forbidden (X), which can be directly compared with ARPES results.

We revised the description of Table S1 accordingly and replaced the triangular symbols with circular symbols.

2. According to Figure 2, the electron band at Gamma has a very low coherence temperature. If, as the authors argued, this electron band is Te 5p_z band, why does Te 5p_z band have such low coherence temperature? The argument that it has some small mixture of Fe 3d_{xy} orbital is not convincing.

The reviewer raised a question about the origin of the unexpectedly low coherence temperature of the Te 5p_z band with a mixture of the Fe 3d_{xy} orbital. We would like to provide more detailed explanations.

The Te 5p_z electron band observed by ARPES in this work is expected to be the antibonding state of Te 5p_z orbital and d_{xy} orbital according to previous study results^{1,2}, implying a partial mixture with d_{xy} orbitals. In addition, an orbital-selective Mott phase was proposed for the corresponding band, which implies a significantly low coherence temperature². Therefore, the Te 5p_z band has a sizable mixture of d_{xy} orbital, which naturally leads to a low coherence temperature.

3. The hybridization gap. The authors argued that they observed a hybridization gap between a Fe 3d_{xy} band and the Te 5p_z band along Gamma-Z path as illustrated in Figure 5. However, according to symmetry and DFT calculations, the crossing of the Fe 3d_{xy} band and Te 5p_z band is protected by symmetry, i.e., there should be no hybridization gap opened between these two

bands around the crossing point. The authors should give good reasons why these two bands hybridize and open a gap (what breaks the symmetry protection?)

As the reviewer correctly pointed out, the hybridization between d_{xy} and p_z band is protected by crystal C_4 rotational symmetry even after the inclusion of spin-orbit coupling³. However, when FeTe enters the bicollinear antiferromagnetic state, the C_4 rotational symmetry is broken because the magnetic transition is accompanied by a structural monoclinic transition and the magnetism itself is C_2 symmetric⁴⁻⁶. This broken symmetry allows hybridization between d_{xy} and p_z orbital. The lowered rotational symmetry of the electronic structure is further evidenced by the resistivity anisotropy of FeTe in the antiferromagnetic state^{4,7}. We would like to point out that the DFT calculation results shown in the Supplementary Information are performed in the *paramagnetic state*, which does not take the symmetry lowering into account.

4. If the electron band at Gamma is Te 5p_z band, it means this Te 5p_z band is below the Fermi level at Gamma point and the entire Te 5p_z band is below the Fermi level along the Gamma-Z path, which contradicts with Figure 5 a.b. and Figure. S9. Therefore, there is no crossing between the Fe 3d_xy band and Te 5p_z band along the Gamma-Z path. As a result, no hybridization gap can open between the Fe 3d_xy band and Te 5p_z band along the Gamma-Z path. This contradicts with the claim of the manuscript.

The reviewer argues that the hybridization gap should not be observed if the electron band is at the Γ point. However, we wish to point out that the electron band crosses the Fermi level at an intermediate point between Γ and Z as illustrated in Fig. 5 (denoted as Γ'). The k_z value of the Γ' point can be approximately determined as $k_z \approx 0.5 [\pi/c]$ with an inner potential of 13 eV⁸. We do not claim that the electron band crosses the Fermi level at the true Γ point and none of the experimental observations implies the crossing of the Fermi level at the Γ point. The reviewer's misunderstanding might come from the conventional notation used in this work (also commonly used in the community); we described that the electron is measured near the Γ point, but the Γ point was to represent the center point of the in-plane Brillouin zone, not the true 3D Brillouin zone. In this respect, we realized that the terminology used in this work and the reply should be more clearly defined. So, we added apostrophes to the Brillouin zone symbols if needed (e.g. Γ') and added the definition of the k_z position of the Γ' to the main manuscript.

Regarding band folding, while the bicollinear antiferromagnetic state (BAFM) is the ground state of bulk FeTe, the stripe ($\pi,0$) antiferromagnetic state (SAFM) is very close to the BAFM state in energy, i.e., the SAFM state competes strongly with the BAFM state. Due to material defects and finite temperature effect, I believe there are band folding from the M point to the Gamma point in the ARPES experiments due to partial SAFM order in the FeTe sample.

If partial SAFM order is responsible for the band folding, the order (or related fluctuations) should be well defined in the momentum space. A previous neutron scattering study on $\text{FeTe}_{1-x}\text{Se}_x$ reported that the SAFM fluctuations are significantly suppressed in the BAFM regime. In addition, a recent elastoresistivity study on $\text{FeTe}_{1-x}\text{Se}_x$ reported that there is no sizable B_{2g} channel (related to SAFM) signal near the FeTe end whereas a strong B_{1g} channel (related to BAFM) signal is observed. It also implies that SAFM order or related fluctuation is very weak in FeTe. Considering these results, we respectfully disagree with the reviewer that the SAFM order or related fluctuation

induced the band folding from the M' point.

References

- 1 Wang, Z. *et al.* Topological nature of the FeSe_{0.5}Te_{0.5} superconductor. *Physical Review B* **92**, doi:10.1103/PhysRevB.92.115119 (2015).
- 2 Kim, M., Choi, S., Brito, W. H. & Kotliar, G. Orbital Selective Mott Transition Effects and Non-Trivial Topology of Iron Chalcogenide. *arXiv preprint arXiv:2304.05002* (2023).
- 3 Zhang, P. *et al.* Multiple topological states in iron-based superconductors. *Nature Physics* **15**, 41-47, doi:10.1038/s41567-018-0280-z (2018).
- 4 Jiang, J. *et al.* Distinct in-plane resistivity anisotropy in a detwinned FeTe single crystal: Evidence for a Hund's metal. *Physical Review B* **88**, doi:10.1103/PhysRevB.88.115130 (2013).
- 5 Mizuguchi, Y., Tomioka, F., Tsuda, S., Yamaguchi, T. & Takano, Y. FeTe as a candidate material for new iron-based superconductor. *Physica C: Superconductivity* **469**, 1027-1029, doi:10.1016/j.physc.2009.05.177 (2009).
- 6 Bishop, C. B., Moreo, A. & Dagotto, E. Bicollinear Antiferromagnetic Order, Monoclinic Distortion, and Reversed Resistivity Anisotropy in FeTe as a Result of Spin-Lattice Coupling. *Phys Rev Lett* **117**, 117201, doi:10.1103/PhysRevLett.117.117201 (2016).
- 7 Kim, Y., Huh, S., Kim, J., Choi, Y. & Kim, C. Magnetic field detwinning in FeTe. *Progress in Superconductivity and Cryogenics (PSAC)* **21**, 6-8 (2019).
- 8 Lohani, H. *et al.* Band inversion and topology of the bulk electronic structure in FeSe_{0.45}Te_{0.55}. *Physical Review B* **101**, 245146 (2020).